# Decoding coral resistance to eutrophication through the association of hyper-efficient denitrifiers as key microbial allies

Nan Xiang [1,7], Tianhua Liao [1,7], Mei Xie[1], Ziyan Wang [2], Chun Ho Mak [2], Xiaowei Tang[2], Shelby E. McIlroy[1,2], Benoit Thibodeau [2,3], Christian R. Voolstra [4] & Haiwei Luo [1,2,3,5,6]

Coral reefs face a perilous future due to global climate change compounded by the increasing prevalence of local stressors. Prominent among these is nutrient pollution, particularly nitrate eutrophication, which disrupts the coral-algal symbiosis and escalates reef degradation. While microbial denitrification is hypothesized to mitigate nitrate stress, the mechanisms underlying coral resilience remain unknown. Studying Hong Kong's coral "reef oases" that persist under chronic hyper-eutrophication, we discovered that resilience is not mediated by diversity or abundance shifts in denitrifier genera but by the association with specific, hyper-efficient denitrifying populations within the dominant denitrifier genus *Ruegeria*. By integrating population genomics, subspecies-resolution metabarcoding (resolving both the entire *Ruegeria* community and the denitrifying sub-community), and direct isotope-based activity assays, we identified and validated putative denitrifying "specialist" populations. These specialists were significantly enriched in corals from high-nitrate waters and exhibited 10-fold higher denitrification rates in low-oxygen incubations, converting nitrate to inert $N_2$ with superior efficiency compared to non-specialists. Our findings reveal that critical ecosystem-scale adaptations to anthropogenic change can occur through a unique association with specialized sub-genus populations, which may be missed in conventional microbiome surveys. As such, our work sheds light into why dominant denitrifying genera are ubiquitous, yet only certain corals thrive in eutrophic conditions. It also provides a framework for future studies delineating ecologically important host-associated microbes.

Coral reefs face threats from anthropogenic stressors, with rising ocean temperatures and coastal pollution pushing these ecosystems toward collapse[1,2]. While heat-induced coral bleaching has dominated scientific literature[3], another threat, namely nutrient pollution, is increasingly recognized as a critical driver of reef degradation[4,5]. Nitrate enrichment on reefs is mainly sourced from industrial sewage and agriculture[4]. It disrupts the carbon to nitrogen stoichiometry by stimulating symbiotic algal Symbiodiniaceae proliferation, reducing

[1]Simon F. S. Li Marine Science Laboratory, School of Life Sciences, The Chinese University of Hong Kong, Shatin, Hong Kong SAR. [2]School of Life Sciences, The Chinese University of Hong Kong, Shatin, Hong Kong SAR. [3]Department of Earth and Environmental Sciences, The Chinese University of Hong Kong, Shatin, Hong Kong SAR. [4]Department of Biology, University of Konstanz, Konstanz, Germany. [5]State Key Laboratory of Agrobiotechnology, The Chinese University of Hong Kong, Shatin, Hong Kong SAR. [6]Institute of Environment, Energy and Sustainability, The Chinese University of Hong Kong, Shatin, Hong Kong SAR. [7]These authors contributed equally: Nan Xiang, Tianhua Liao. ✉e-mail: haiweiluo@cuhk.edu.hk

carbon translocation to the coral host[6,7], and eventually leading to symbiotic breakdown[8]. Nitrate enrichment induces holobiont oxidative stress that restricts coral growth and reproduction[9], and interferes with biomineralization, leading to reduced calcification rates[5]. Excess nitrate also interacts synergistically with warming to lower coral thermal tolerance[9]. At the reef scale, nitrate enrichment fuels macroalgal overgrowth[10] and microbial destabilization[11], further shifting healthy reefs toward degradation states[2].

Denitrifiers, a group of microbes that can enzymatically reduce nitrate/nitrite to nitrous oxide ($N_2O$) or dinitrogen ($N_2$), are considered key players in coral holobiont functioning, particularly on nutrient-rich reefs[12,13]. Unlike well-established $N_2$ fixation by diazotrophs[14–17], denitrification has rarely been investigated in coral holobionts[15]. Denitrifiers are prevalent in corals, with their abundance and activity positively correlating with coral autotrophic capability[18,19], a relationship likely driven by the supply of organic carbon from algal photosynthesis to fuel denitrification in coral holobionts. Putative denitrifiers in octocorals *Xenia umbellata* and *Pinnigorgia flava* in an indoor aquarium facility were predominantly composed of Rhodobacterales members, including *Ruegeria*, *Dinoroseobacter*, and *Labrenzia*[20]. Given appropriate environmental conditions (i.e., low oxygen but adequate organic carbon and nitrate availability), microbial denitrification potential is largely shaped by evolutionary pressures, including gene gain and loss[21], regulatory machine adaptations that fine-tune enzyme expression[22], and metabolic flexibility constrained by redox regimes[23]. For instance, lineages within the denitrifier genus *Pseudomonas* harbor distinct clusters of modular denitrification genes due to gene gain and loss, which resulted in their varying denitrification activities under nitrate spikes[24].

Nitrate eutrophication is a pervasive issue in coastal reef persistency globally[25,26], with concentrations exceeding natural thresholds by orders of magnitude in regions like the Great Barrier Reef[27], Caribbean[10], Southeast Asia[28], Coral triangle[29], and notably, Hong Kong's heavily urbanized reefs[30,31]. Despite intense regional urbanization, Hong Kong boasts high coral biodiversity with over eighty coral species inhabiting the reefs[30]. Yet, intensive human activities subjected Hong Kong reefs to severe nutrient pollution[31], with nitrate concentrations consistently exceeding $4\,\mu M$ and reaching peaks of up to $20\,\mu M$, far above the $1–2\,\mu M$ threshold known to disrupt coral physiology[32]. This resilience to high nutrients positions Hong Kong's eutrophic reefs as a model "coral reef oasis", a category of marginal refuges distinct from thermally resilient "super reefs"[33].

The persistence of these "coral reef oases" under hypereutrophication conditions points to a putative adaptive mechanism: corals may enlist microbial allies to mitigate nitrate stress. This potential for microbiome-mediated resilience, particularly at the fine-scale microbial population (i.e., sub-genus) level, remains largely unexplored. Hong Kong's reefs, with their strong west-to-east nitrate gradient year-round (i.e., during both dry and wet seasons)[31] (Fig. 1a and Supplementary Data 1), provide an ideal natural experiment to test this. Here, we show that corals thriving in chronically eutrophic western Hong Kong waters are associated with specialized denitrifying populations within the ubiquitous bacterial genus *Ruegeria*[20,34]. These populations possess high denitrification activities, putatively transforming a nitrate burden into a metabolic detoxification mechanism in natural coral reefs. Our work provides insights on the interplay between microbial evolution and coral holobiont adaptation to the natural nitrate gradients using a multidisciplinary approach (Supplementary Fig. 1).

## Results

### *Ruegeria* dominate the coral holobiont denitrifying community across a nitrate gradient

To assess the overall structure of the coral-associated denitrifier community along nitrate gradients, we performed genus-level profiling by conducting amplicon sequencing of a denitrification marker gene. This approach allowed us to test whether local water nitrate gradients shape the denitrifying community compositions at the taxonomic level of bacterial genera. The nitrite reductase genes, *nirS* and *nirK* are widely used as functional markers to assess the diversity of denitrifiers, as they encode nitrite reductases critical for the denitrification process[35]. These genes are highly conserved among denitrifiers, making them reliable indicators of denitrifier communities[36,37]. Here, we targeted the *nirS* gene, given its greater prevalence in marine ecosystems[38] and its validated usage in characterizing denitrifier communities in corals[19,20]. The *nirS* amplicon analysis revealed that denitrifier abundances in corals (i.e., using combined sequenced compartments of mucus, tissue, and the skeleton) were evenly distributed at the genus level across the nitrate gradient (Fig. 1ab; PERMANOVA, site effect, $F_{6,67} = 46.24$, $P = 0.201$; coral effect, $F_{4,67} = 29.22$, $P = 0.232$). At the genus level, denitrifier communities in Hong Kong corals were largely similar across coral species and sampling sites, with *Ruegeria* (~12%), *Pseudomonas* (~12%), *Jhaorihella* (~10%), and *Halomonas* (~7%) dominating (Fig. 1b). The high abundance of *Ruegeria* further confirms its suggested importance in coral-associated denitrification.

### A majority of coral-associated *Ruegeria* isolates carry a complete denitrification gene set

Having characterized the overall denitrifier community in corals, we next cultured isolates belonging to the predominant genus *Ruegeria,* with their denitrification potential genomically evaluated. We obtained 419 *Ruegeria* isolates from five coral species across nine Hong Kong reef sites, with 326 isolates from *Oulastrea crispata*, 2 from *Acropora* cf. *samoensis*, 43 from *Acropora solitaryensis*, 29 from *Platygyra acuta*, and the remaining 19 from *Favites pentagona*, respectively (Fig. 1c; Supplementary Data 2 and Data 3). A phylogenomic tree of *Ruegeria* isolates, constructed using a core-genome alignment of these isolates, along with 26 publicly available *Ruegeria* genomes as the reference (Fig. 1c and Supplementary Data 4), was used to depict the distribution of key denitrification genes across isolates within this genus. Of these, 355 isolates (~84.72% of total) were genetically equipped with the complete denitrification pathway, encoding *napA* (340/355) or *narG* (15/355) for nitrate reductase, *nirS* (355/355) or *nirK* (1/355) for nitrite reductase, *norB* (355/355) for nitric oxide reductase, and *nosZ* (355/355) for nitrous oxide reductase (Fig. 1c and Supplementary Data 3). The remaining 64 isolates either possessed partial denitrification gene sets ($n = 50$) or lacked any denitrification genes ($n = 14$) (Supplementary Data 3). Genome completeness was uniformly high across all isolates ($98.59 \pm 1.31\,\%$), with no significant difference (Mann–Whitney $U$ test, $P = 0.901$) between isolates carrying complete ($98.58 \pm 1.41\,\%$) versus partial or no denitrification genes ($98.66 \pm 0.55\,\%$), confirming that this variation reflects biological differences rather than sequencing or assembly artifacts.

### Nitrate gradient structures *Ruegeria* at the population level, not the genus level

The apparent lack of enrichments of dominant denitrifying genera like *Ruegeria*, *Pseudomonas*, *Jhaorihella* in high-nitrate western Hong Kong waters (Fig. 2ab), contrasted by the enrichment of rare taxa such as *Thiobacillus* and *Idiomarina* (Fig. 2ab), presents a paradox in understanding denitrification in eutrophic coral reefs. Given that these dominant denitrifying genera are likely key players in removing excess nitrate/nitrite for corals in eutrophic reefs, we hypothesized that adaptive differentiation might occur at the sub-genus level. To test this, we sought to determine whether specific, ecologically distinct populations within these genera were in fact being selected for in high-nitrate environments.

We therefore employed a population-resolved approach to dissect the *Ruegeria* community. We followed a population genetics

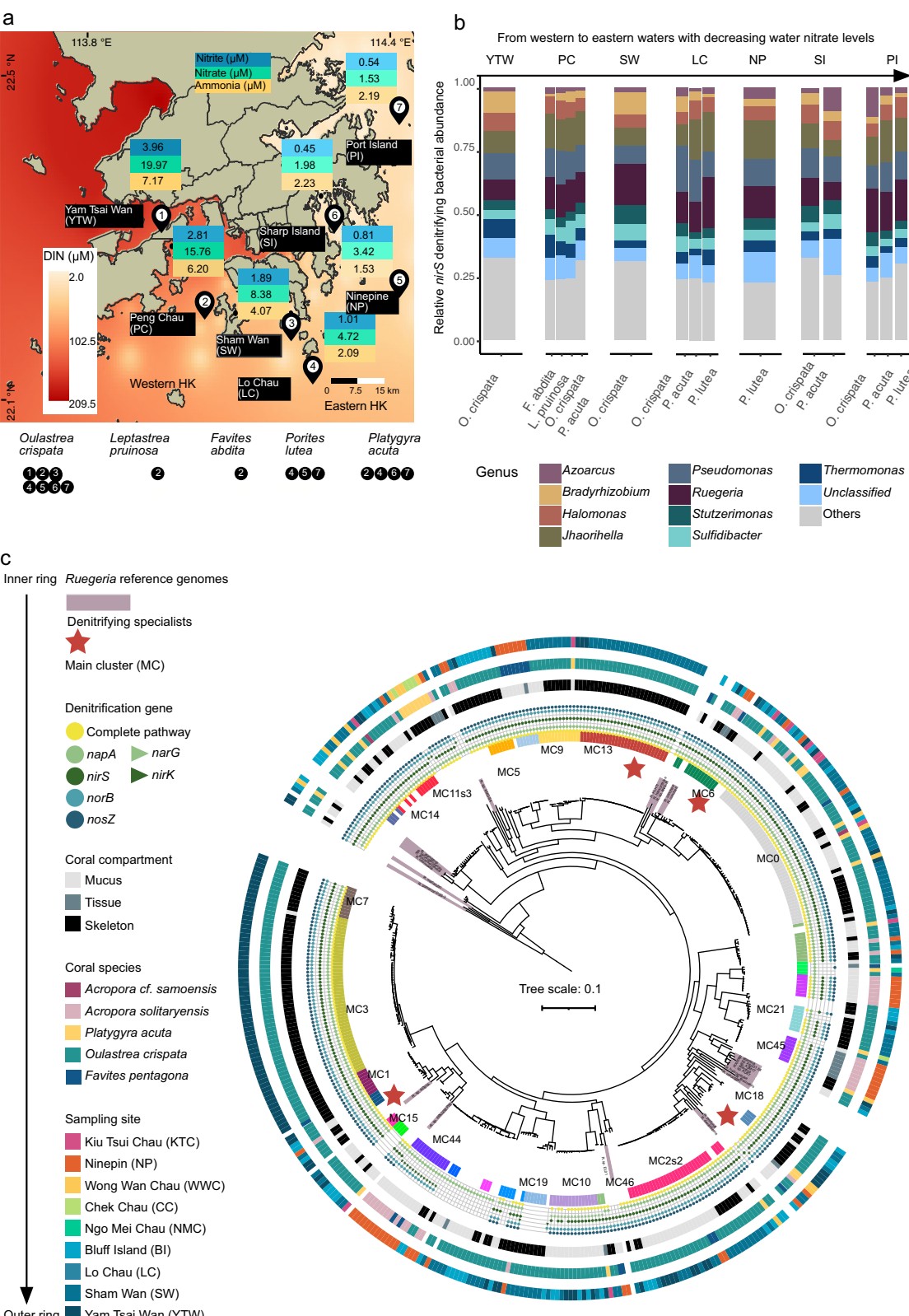

**a**

Nitrite (µM)
Nitrate (µM)
Ammonia (µM)

Yam Tsai Wan (YTW): 3.96 / 19.97 / 7.17
Peng Chau (PC): 2.81 / 15.76 / 6.20
Port Island (PI): 0.54 / 1.53 / 2.19
Sharp Island (SI): 0.45 / 1.98 / 2.23
Sham Wan (SW): 1.89 / 8.38 / 4.07
Ninepine (NP): 0.81 / 3.42 / 1.53
Lo Chau (LC): 1.01 / 4.72 / 2.09

DIN (µM): 2.0 / 102.5 / 209.5

Western HK    Eastern HK

*Oulastrea crispata* ① ② ③ ④ ⑤ ⑥ ⑦
*Leptastrea pruinosa* ②
*Favites abdita* ②
*Porites lutea* ④ ⑤ ⑦
*Platygyra acuta* ② ④ ⑥ ⑦

**b** From western to eastern waters with decreasing water nitrate levels

Relative *nirS* denitrifying bacterial abundance

**Genus**
- *Azoarcus*
- *Bradyrhizobium*
- *Halomonas*
- *Jhaorihella*
- *Pseudomonas*
- *Ruegeria*
- *Stutzerimonas*
- *Sulfidibacter*
- *Thermomonas*
- *Unclassified*
- Others

**c**

Inner ring — *Ruegeria* reference genomes

Denitrifying specialists ★

Main cluster (MC)

**Denitrification gene**
- Complete pathway
- *napA*
- *nirS*
- *norB*
- *nosZ*
- *narG*
- *nirK*

**Coral compartment**
- Mucus
- Tissue
- Skeleton

**Coral species**
- *Acropora cf. samoensis*
- *Acropora solitaryensis*
- *Platygyra acuta*
- *Oulastrea crispata*
- *Favites pentagona*

**Sampling site**
- Kiu Tsui Chau (KTC)
- Ninepin (NP)
- Wong Wan Chau (WWC)
- Chek Chau (CC)
- Ngo Mei Chau (NMC)
- Bluff Island (BI)
- Lo Chau (LC)
- Sham Wan (SW)
- Yam Tsai Wan (YTW)

Outer ring

Tree scale: 0.1

framework, where a "population" is defined as a group of individuals of the same species that exchange genetic material and inhabit a specific geographic area, thereby sharing a common gene pool. The PopCO-GenT tool[39] delineates such populations by identifying clusters of recent gene flow, which represent ecologically cohesive and genetically isolated units. As all our isolates originate from Hong Kong's corals, the populations identified conform to this definition: they are

sub-genus groups from a specific geographic area among which recent gene exchange has been significantly more frequent than with other groups. Using this tool, 334 of the 419 *Ruegeria* isolates were classified into 29 populations (i.e., main clusters, MCs; each MC containing at least three isolates; Supplementary Data 2). Genomes within the same *Ruegeria* MC showed an average nucleotide identity (ANI) of 98.91 ± 0.84%, exceeding the established operational species

**Fig. 1 | Prevalence and genomic characteristics of denitrifying bacteria in corals from Hong Kong reefs. a** Locations of seven Hong Kong reef sites. Shown are the calculated decadal mean dissolved inorganic nitrogen (DIN) concentrations as well as nitrate, nitrite, and ammonia levels in waters across sites. Coral species sampled at each site were indicated in white, encircled numbers. These species were chosen given their consistent prevalence from western to eastern reefs with water DIN gradients. Data were retrieved from the monthly measurements of 76 seawater quality monitoring stations during the years 2012–2022, published by the Agriculture, Fisheries, and Conservation Department of Hong Kong. **b** Relative abundance of *nirS*-harboring denitrifiers in five coral species across seven sites, revealing the high abundance of *Ruegeria* in coral-associated denitrifiers. The stacked bar plot showing denitrifier community compositions in corals, with 10 most abundant bacterial genera displayed. Abbreviations for sampling sites were: YTW for Yam Tsai Wan, PC for Peng Chau, SW for Sham Wan, LC for Lo Chau, NP for Ninepin, SI for Sharp Island, and PI for Port Island. **c** Rooted maximum-likelihood

phylogenomic tree of 419 *Ruegeria* isolates obtained from five coral species across nine Hong Kong reef sites, along with 26 *Ruegeria* genomes retrieved from public databases. This tree was constructed with IQ-TREE based on concatenated single-copy orthologous genes at the amino acid level. A total of 19 *Ruegeria* populations (i.e., main clusters, MCs) that contain representative members additionally sequenced by Nanopore were indicated in different colors in the tree. MCs were delineated using PopCOGenT. From the inner to the outer rings: reference genomes, denitrifying specialists (labelled with solid red stars), main cluster (MC), denitrification gene, coral compartment, coral species, and sampling site. Denitrification panel included a complete pathway labelled with solid yellow circles and key gene presence, i.e., *napA/narG* for nitrate reductase, *nirS/nirK* for nitrite reductase, *norB* for nitric oxide reductase, and *nosZ* for nitrous oxide reductase. More than 80% of *Ruegeria* isolates were equipped with genes capable of the complete denitrification pathway. Key denitrification steps were shown in different colors with different solid symbols indicating alleles for the same step.

threshold of 95%. ANI values between *Ruegeria* MCs ranged from 86.85% to 96.95%. To track these populations in environmental samples, we utilized the single-copy, essential genes *ATP5B* (encoding a subunit of mitochondrial ATP synthase) and *parC* (encoding DNA topoisomerase IV subunit A) as robust taxonomic population markers. Their universal presence provides a phylogenetic signal to resolve and quantify the total *Ruegeria* community structure. In parallel, we used the *nirS* gene as a functional population marker; as a key denitrification enzyme not universal in *Ruegeria*, it specifically resolves the denitrifying *Ruegeria* sub-community. Phylogenetic trees constructed from amplified regions of all three genes confirmed their utility by grouping *Ruegeria* MCs into distinct clusters, validating their power to resolve populations (Fig. 2c).

Non-metric multidimensional scaling (NMDS) ordinations based on the taxonomic markers revealed that overall *Ruegeria* population compositions were distinct across sites but not coral species (Fig. 2d). Crucially, analysis of the population-resolving *nirS* marker showed that the compositions of denitrifying *Ruegeria* populations varied significantly across sites, with a clear separation between the two high-nitrate western sites (i.e., Yam Tsai Wan and Peng Chau) and all others (i.e., Sham Wan, Lo Chau, Ninepin, Sharp Island, and Port Island). This stark contrast between the site-specific denitrifying *Ruegeria* community composition and the evenly distributed denitrifying genera across sites (Fig. 2b, d) highlights the critical population dynamics we aimed to uncover. ASVs were assigned to their respective MCs by aligning ASVs to the corresponding gene sequences extracted from *Ruegeria* genomes. Envfit analysis on amplicon profiles identified several MCs, namely MC1 ($ATP5B$, $R^2 = 0.039$, $P = 0.291$; $parC$, $R^2 = 0.676$, $P = 0.001$; $nirS$, $R^2 = 0.146$, $P = 0.001$), MC6 ($ATP5B$, $R^2 = 0.749$, $P = 0.001$; $parC$, $R^2 = 0.636$, $P = 0.001$; $nirS$, $R^2 = 0.546$, $P = 0.001$), MC13 ($ATP5B$, $R^2 = 0.516$, $P = 0.001$; $parC$, $R^2 = 0.532$, $P = 0.001$; $nirS$, $R^2 = 0.529$, $P = 0.001$), and MC18 ($ATP5B$, $R^2 = 0.227$, $P = 0.001$; $parC$, $R^2 = 0.101$, $P = 0.021$; $nirS$, $R^2 = 0.292$, $P = 0.001$), that were concomitantly enriched in high-nitrate western waters in both the entire *Ruegeria* community (based on $ATP5B$ and/or $parC$, Fig. 2d and Supplementary Data 5) and the denitrifying sub-community (based on $nirS$, Fig. 2d and Supplementary Data 5). We named these populations as "denitrifying specialists". Notably, available isolates of denitrifying specialists were all sourced from the coral species *Oulastrea crispata*, with many of them being isolated from the skeleton, except one, which was obtained from the mucus layer.

Those denitrifying specialist MCs made up 1-10% of coral-associated *Ruegeria* communities in two high-nitrate western reefs, but only 0.1-2% in non-western reefs (Fig. 2e). Although MC7 and MC40 showed consistent enrichments in both total and denitrifying *Ruegeria* communities of western water corals (Fig. 2d), they were not considered as the "denitrifying specialists" as each constituted <1% of the total *Ruegeria* populations (Supplementary

Fig. 2), below our abundance threshold (>1%) for specialist classification. The Envfit analysis mapping seawater physiochemical parameters with *Ruegeria* population compositions identified total phosphorus ($ATP5B$, $R^2 = 0.819$, $P = 0.001$; $parC$, $R^2 = 0.817$, $P = 0.001$; $nirS$, $R^2 = 0.941$, $P = 0.001$) and nitrate ($ATP5B$, $R^2 = 0.804$, $P = 0.001$; $parC$, $R^2 = 0.810$, $P = 0.001$; $nirS$, $R^2 = 0.899$, $P = 0.001$) as top two driving forces for denitrifying specialist MC enrichments in western water corals, followed by nitrite and ammonia (Fig. 2f and Supplementary Data 6).

### Denitrifying specialists exhibit hyper-efficient N₂ production

To determine whether the denitrifying specialists exhibit superior denitrification activity than non-specialists, we quantified their productions of $N_2O$ and $N_2$ using $^{15}N$-stable isotope labelling. Given that denitrification usually occurs in the anaerobic environment[40], we initially validated our $^{15}N$-stable isotope assay in six *Ruegeria* MCs representing three denitrifying specialists, two denitrifying non-specialists, and a negative control lacking denitrification genes, under suboxic (2.5 µmol l⁻¹ dissolved oxygen, DO) and "nanoxic"[41] conditions (0.5 µmol l⁻¹ DO) (Supplementary Fig. 3), with a more diverse group of denitrifying specialists (four MCs) and non-specialists (nine MCs) assayed under the nanoxic condition. The phylogenomic tree of the 29 assayed *Ruegeria* isolates (Fig. 3a and Supplementary Data 7) was used for the phylANOVA (Phylogenetic ANOVA) analysis, which controls for an evolutionary dependence of isolates during the comparisons of denitrification activity between specialist vs. non-specialist MCs. Denitrifying specialist *Ruegeria* MCs exhibited significantly higher productions of $^{46}N_2O$ compared to their non-specialist counterparts (PhylANOVA, $P = 0.014$; Fig. 3ab; Supplementary Data 8) under the nanoxic condition. Scattered positions of specialist MCs on the phylogeny (Fig. 1 and 3a) indicated that their denitrification capabilities evolved convergently across distinct lineages. The highest $^{46}N_2O$ production was observed in the denitrifying specialist MC6, specifically in the isolates MC6-AA3 (1.34 pmoles ml⁻¹ h⁻¹) and MC6-O11 (0.71 pmoles ml⁻¹ h⁻¹) (Fig. 3b and Supplementary Data 9). Similarly, denitrifying specialist *Ruegeria* MCs produced significantly higher $^{30}N_2$ than their non-specialist counterparts (PhylANOVA, $P = 0.015$), pronouncedly reflected in MC1 and MC6 under the nanoxic condition (Fig. 3a, c and Supplementary Data 8). The top four highest $^{30}N_2$ productions were found in four isolates belonging to four different denitrifying specialist MCs, i.e., MC1-AF10 (472.5 pmoles ml⁻¹ h⁻¹), MC6-O11 (212.30 pmoles ml⁻¹ h⁻¹), MC13-L12 (165.88 pmoles ml⁻¹ h⁻¹), and MC18-BF4 (160.28 pmoles ml⁻¹ h⁻¹) (Fig. 3c and Supplementary Data 9). It is worth mentioning that we search for denitrifying specialists based on metabarcoding analysis at the population level, but the isolates affiliated with specialist MCs were not necessarily sourced from the western sites only (Fig. 3a; Supplementary Data 7). To test the correlation between the geographical origin of isolates (western vs. non-western

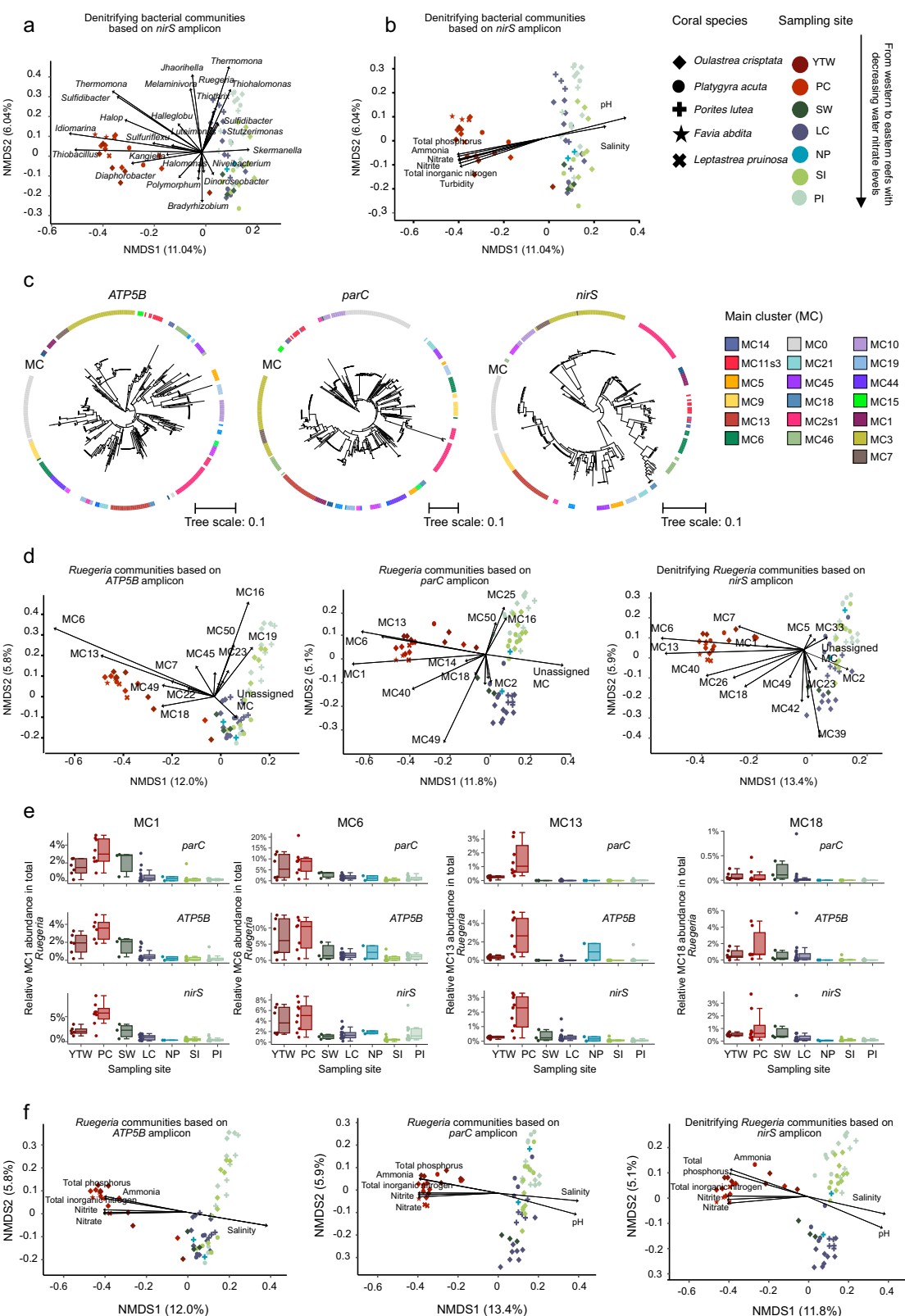

f

sites) and their denitrification activity phenotypes, we performed chi-square tests, which are suitable for analysing associations between categorical variables. *Ruegeria* isolation site and denitrification activity showed statistically significant associations (chi-square test, $^{30}N_2$, $P = 4.93e\text{-}06$; $^{46}N_2O$, $P = 0.025$), with western sourced *Ruegeria* isolates exhibiting higher denitrification activities than non-western ones. Production of $^{45}N_2O$ and $^{29}N_2$ indicated that *Ruegeria* isolates

incorporated one $^{15}N$ atom from the $^{15}N$-labelled sodium nitrate, paired with one $^{14}N$ atom derived from natural nitrogen sources. Productions of $^{45}N_2O$ and $^{29}N_2$ in all *Ruegeria* isolates were 10- to 20-fold lower than $^{46}N_2O$ and $^{30}N_2$ (i.e., incorporation of two $^{15}N$ atoms from the labelled nitrate), respectively (Fig. 3b, c and Supplementary Fig. 4), confirming the complete utilization of $^{15}N$-labelled sodium nitrate in the denitrification process[42].

**Fig. 2 | Enrichments of denitrifying specialists in corals correlate with water nitrate gradients. a** Non-metric Multidimensional Scaling (NMDS) ordination of *nirS* denitrifier community compositions and Envfit (i.e., fits an environmental vector or factor onto an ordination) showing correlations between coral-associated denitrifying genera distributions and sampling sites. Denitrifying genera with significant correlations (Envfit, $P < 0.05$) to the sites were shown. The significance of envfit correlations was examined using a permutation test (empirical $P$ values based on $R^2$) with a non-directional test of fit. One-sided/two-sided is not applicable. **b** NMDS ordination of denitrifier community compositions and Envfit identifying physicochemical drivers of denitrifying genera enrichments in corals from western waters. **c** Phylogenetic trees for *ATPSB*, *parC*, and *nirS* gene markers demonstrating their ability to resolve *Ruegeria* at the population level. Trees were constructed with IQ-TREE based on the amplified region of each gene at the nucleotide level. *Ruegeria* populations delineated with PopCOGenT were shown in different colors in the trees. **d** NMDS ordination of *Ruegeria* population compositions and Envfit showing correlations between coral-associated *Ruegeria* population distributions and

sampling sites. MCs with significant correlations (Envfit, $P < 0.05$) to the sites were shown. Arrow direction indicates the gradient of increase for a particular variable across the ordination plot, with samples projected in the arrow direction owning high values. Arrow length indicates the correlation strength between a particular variable and *Ruegeria* community composition. **e** Relative abundances of denitrifying specialists in total *Ruegeria* showing their enrichments in corals from high nitrate western waters. *Ruegeria* MCs concomitantly enriched in western water coral-associated *Ruegeria* (based on *ATPSB* or *parC* amplicons) and denitrifying *Ruegeria* (based on *nirS* amplicon) were named as "denitrifying specialists". Box plots were used to display maximum, median, and minimum data with scatterplots depicting the distribution of raw data ($n = 3$–12 for each group). Abbreviations for sampling sites were the same as those in Fig. 1. **f** NMDS ordination of *Ruegeria* population compositions and Envfit identifying total phosphorus and nitrate as two key physicochemical drivers of denitrifying specialist *Ruegeria* enrichments in corals from western waters.

Denitrification genes from denitrifying specialists and non-specialists may have different evolutionary histories. Individual maximum-likelihood gene trees of key denitrification operons (*nap*, *nir*, *nor*, *nos*) were constructed at the amino acid level using the extracted gene sequences from *Ruegeria* genomes. Comparison of these phylogenetic trees and the phylogenomic tree revealed incongruent evolutionary patterns for denitrification genes in denitrifying specialists. Specifically, while MC1 and MC6 are phylogenomically distant, their *nir* and *nos* genes cluster closely together in the gene trees, suggesting convergent evolution possibly through horizontal gene transfer (HGT) at these loci (Supplementary Fig. 5). Despite these differences in gene phylogeny, denitrification operons exhibited highly conserved genomic organization across *Ruegeria* isolates. Gene synteny analysis revealed that denitrification operons were consistently clustered and co-linear in most of the genomes (Supplementary Fig. 6). Operon architecture did not differ between denitrifying specialists and non-specialists (Supplementary Fig. 6). Among eight genomes with circular chromosome (i.e., from eight representative MCs used in the $^{15}N$-labeling stable isotope assay), denitrification genes resided on plasmids in five isolates (specialist: MC1-AL3; non-specialists: MC26-AN11, MC21-BK1, MC3-AL7, MC7-AF3), with *nap, nor, and nos* operons usually located on the same plasmid (Supplementary Fig. 7). The remaining three isolates harbored denitrification genes chromosomally (specialists: MC13-AP11, MC6-AB11; non-specialist: MC0-A5; Supplementary Fig. 7). Notably, the *nap* operon behaved like a mobile genetic cassette, translocating between plasmids or chromosomal loci and displaying variable strand orientation (Supplementary Fig. 7). These results suggest that while denitrification genes in specialists likely undergo distinct evolutionary events, their operon architecture remains highly conserved. These findings cannot fully explain the differential $^{46}N_2O$ and $^{30}N_2$ productions between denitrifying specialist and non-specialist MCs.

**Other genetic correlates with denitrifying specialists**

We further performed comparative genomics to uncover genetic differences, i.e., putative adaptations, underlying the denitrifying specialist phenotype. We used Phylogenetic signal estimation (Phylosig) and Phylogenetic Generalized Linear Mixed Model for Binary data (BinaryPGLMM) analyses[43] to identify orthologous genes differentially associated with specialist MCs, along with the gene tree reconciliation analysis to investigate their evolutionary history (Supplementary Data 10). Gene association analysis based on Phylosig and BinaryPGLMM[43] of gene presence and absence has identified 80 orthologous genes (OGs) that significantly differ between specialists and non-specialists (48 enriched and 32 depleted in specialists; Supplementary Fig. 8 and Supplementary Data 11). Functional annotation revealed that these OGs are involved in diverse metabolic pathways,

suggesting that convergent evolution may have been driven by adaptation to local nutrient gradients through diversifying the metabolic flexibility. Significant metabolic gene replacements were shown between denitrifying specialists and non-specialists, including carbohydrate and xenobiotic metabolism pathways (BinaryPGLMM, $P < 0.05$; e.g., *ppc*, K01595; *mgsA*, K01734; *dad*, K05913; *catB*, K01856; *choD*, K03333; Supplementary Fig. 8 and Supplementary Data 11). Given that envfit identified nitrate and phosphorus as primary environmental drivers of denitrifying specialist enrichments in western waters, we subset nitrogen/phosphorus-related metabolism pathways to uncover the potential adaptive mechanisms. The coordinated decreasing trends in both nitrogen assimilation and phosphonate (C-P bond) acquisition and degradation (e.g., *phn* operon) systems in denitrifying specialists (Supplementary Fig. 8 and Supplementary Data 12) were consistent with a genomic signature of adaptation to nutrient-rich (i.e., excess nitrogen and phosphorus) environments. Specifically, specialists showed a trend of reduced pathways for nitrate/nitrite transporters (BinaryPGLMM, $P = 0.092$, *nrtB/nasE/cynB*, K15577; $P = 0.092$, *nrtA/nasF/cynA*, K15576; $P = 0.103$, *nrtC/nasD*, K15578; Supplementary Fig. 8 and Supplementary Data 12) and assimilatory nitrate/nitrite reductases (BinaryPGLMM, $P = 0.099$, *nasC/nasA*, K00372; $P = 0.103$, *nasE*, K26138; $P = 0.103$, *nasD*, K26139; Supplementary Fig. 8 and Supplementary Data 12). Critically, phosphorus metabolism genes were depleted in denitrifying specialists, including *LHPP* (BinaryPGLMM, $P = 0.038$, phosphatase, K11725), *mdoB* ($P = 0.166$, phosphoglucomutase, K01002), and *phn* operon genes ($P = 0.175$, *phnG*, K06166; $P = 0.231$, *phnH*, K06165; $P = 0.231$, *phnI*, K06164; $P = 0.231$, *phnL*, K05780; Supplementary Fig. 8 and Supplementary Data 12), indicating relaxed selection for phosphonate scavenging in phosphate-rich habitats, a putative key factor in specialist evolution. Conversely, denitrification genes including *nirS* (BinaryPGLMM, $P = 0.251$, K15864), *norE* ($P = 0.126$, K02164), *nos* (*nosD*, $P = 0.243$, K07218; *nosF*, $P = 0.234$, K19340; *nosL*, $P = 0.234$, K19342) trended toward enrichment in denitrifying specialists though were non-significant (Supplementary Fig. 8 and Supplementary Data 12). Reconciliation analysis using ecceTERA and ANGST revealed 92 OGs (including 46 OGs overlapped with those identified by Phylosig and BinaryPGLMM analysis, Supplementary Data 13–14) in denitrifying specialists that likely underwent convergent evolution, possibly through HGT. These included denitrifying nitrous oxide reductase *nosZ* accessory genes (*nosF, nosL*) and phosphorus gradient adaptation genes (*phnY*, K00206, *phnE*, K02042, *nuoA*, K00330, *ccoN*, K00404; Supplementary Data 13–14), suggesting independent acquisition of similar functions under parallel selective pressures. These genes clustered within the same phylogenetic lineages in gene trees (see Code Availability), despite originating from phylogenetically distant MCs. Our findings underscore the role of water nitrogen, and phosphorus gradients in denitrifying specialist niche specialization.

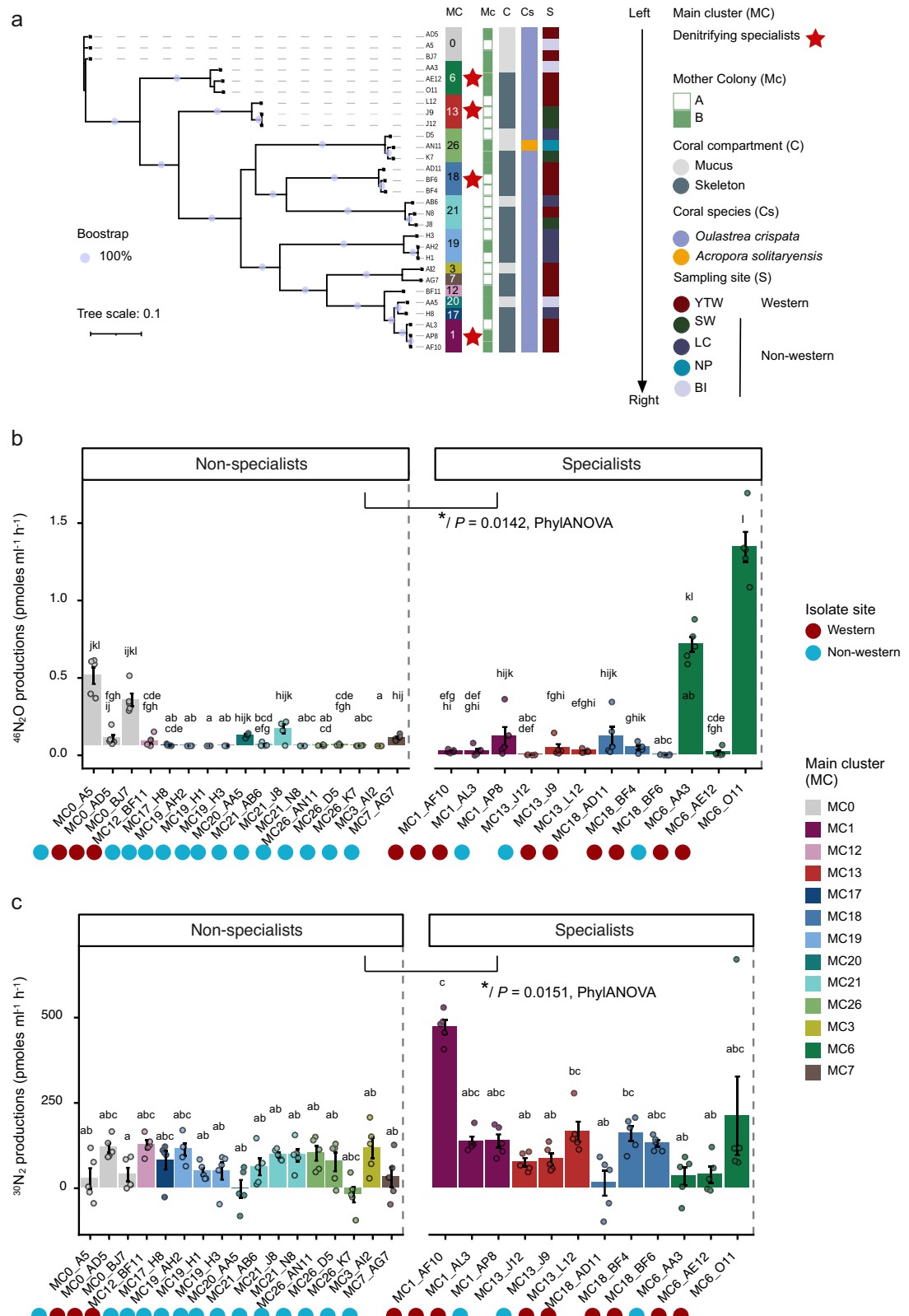

## Discussion

*Ruegeria* exemplifies bacterial genera exhibiting profound intrageneric functional heterogeneity, where ecologically relevant adaptations operate at the population level[44–46]. For example, *Ruegeria* populations are distinct among different coral compartments with varying metabolic capabilities, particularly in carbon and sulfur cycling, which are directly linked to ecological niche adaptation[44].

Here, we provide evidence extending this population-level paradigm to a natural environmental stressor, suggesting that coral resilience to chronic nitrate pollution is hard to infer from diversity or abundance shifts in denitrifier genera, but rather driven by the association with specific, hyper-efficient denitrifying populations within the dominant genus *Ruegeria*. By examining natural populations across a steep nutrient gradient, we show that denitrifying specialist populations (i.e.,

**Fig. 3 | Superior denitrification activities in denitrifying specialists compared to their non-specialist counterparts. a** Rooted maximum-likelihood phylogenomic tree of 29 *Ruegeria* isolates (representing 13 MCs, three isolates for each specialist MC, one to three isolates for each non-specialist MC) used in the $^{15}$N-stable isotope assay. This tree was constructed with IQ-TREE based on concatenated single-copy orthologous genes at the amino acid level and used for PhylANOVA analysis (a phylogenetic ANOVA accounting for evolutionary dependence across lineages). Solid purple circles in the phylogenomic tree indicated nodes with bootstrap support of 100%. From the left to the right: Main cluster (MC), denitrifying specialists (labelled with solid red stars), mother colony (Mc), Coral compartment (C), Coral species (Cs), and sampling site (S). *Ruegeria* populations delineated with PopCOGenT were shown in different colors in the tree. Abbreviations for sampling sites were: YTW for Yam Tsai Wan, SW for Sham Wan, LC for Lo Chau, NP for Ninepin, and BI for Bluff Island. YTW was a western site, and the remaining were all

non-western sites. **b** Superior $^{46}$N$_2$O and **c** $^{30}$N$_2$ production rates in denitrifying *Ruegeria* specialists (four MCs) compared to non-specialist counterparts (9 MCs) under the ecologically relevant nanoxic (0.5 µmol l$^1$ dissolved oxygen) condition. Data were shown with the mean values and standard errors ($n$ = 5 biological replicates each), with *Y*-axis scale in $^{30}$N$_2$ productions two orders of magnitude higher than in $^{46}$N$_2$O productions. Scatterplots depict the distribution of raw data ($n$ = 5 for each group). Coral sampling sites for *Ruegeria* isolates were indicated in solid blue circles for non-western sites (i.e., SW, LC, NP, and BI) and solid red circles for a western site (i.e., YTW). Asterisks denoted significant differences (PhylANOVA with Brownian motion evolution model, $P < 0.05$, *) in $^{46}$N$_2$O ($P = 0.0142$) and $^{30}$N$_2$ productions ($P = 0.0151$) between denitrifying specialists versus non-specialists. Lowercase letters indicated significant differences (two-way ANOVA with Tukey's HSD, $P < 0.05$) between individual *Ruegeria* isolates. Each pair of groups was compared using a two-tailed $P$ value ($n$ = 5 for each test).

MC1, MC6, MC13, and MC18) each constituted up to 2–10% of the total *Ruegeria* community in corals from high-nitrate western waters, a stark contrast to their near absence in samples from non-western, lower nitrate reefs. Denitrifying specialists demonstrated remarkably higher denitrification activities than non-specialists under ecologically relevant "nanoxic" conditions (-0.5 µmol l$^{-1}$ O$_2$)[41]. As such, the prevalence of denitrifying specialists in corals from high-nitrate waters implies that hosts may possess the capacity for associations of microbial allies at a sub-genus population level to support their survival in eutrophicated reefs, though other processes, such as environmental filtering and inter-microbial competition, cannot be ruled out.

To date, most microbiome studies rely on genus/species-level profiling, overlooking critical functional stratifications at the finer population level[47]. Combining amplicon sequencing, population genomics, and direct quantification of denitrification activity via $^{15}$N-labeling stable isotope assays, we demonstrate that denitrifying specialists exhibited -10-fold higher N$_2$ production rates than non-specialists. This genotype-phenotype linkage is notable given the potential decoupling at the transcriptional/translation level or through post-transcriptional[48] and post-translational modifications[49]. Specialist MCs are defined by their collective environmental enrichment in high nitrate waters with intra-population functional differences that can arise through migration and relaxed selection. Members of a specialist population can migrate to low-nitrate sites (e.g., eastern reefs), where selective pressure for high denitrification activity may be relaxed. Over time, genetic drift or local adaptation could lead to reduced denitrification efficiency in some individuals, yet these isolates remain part of the same gene-flow unit (i.e., the same population) and retain the genomic backbone that defines the MC. Denitrifying specialist isolates MC1-AF0 and MC6-O11 achieved exceptional N$_2$ production rates (-480 and 210 pmol N$_2$ ml$^{-1}$ h$^{-1}$, respectively), implying their exceptional nitrogen removal potential in coral holobionts. To our knowledge, this study pioneered the quantification of complete denitrification rates (considering both N$_2$O and N$_2$) for pure denitrifier cultures isolated from corals using the $^{15}$N-stable isotope assay. Direct and simultaneous quantification of both $^{15}$N-labelled N$_2$O and N$_2$[50] represents a methodological advancement over traditional approaches that infer denitrification rates solely from N$_2$O accumulations[51,52], enabling accurate resolution of denitrification activity in specialist and non-specialist MCs. Importantly, our identification of denitrifying specialists is based on population-level metabarcoding (using *parC/ATP5B* markers for total *Ruegeria* community and *nirS* markers for denitrifying *Ruegeria* sub-community), revealing their significant enrichment in high-nitrate reefs, even though individual isolates affiliated with these specialist populations can be sourced from other, lower-nitrate sites. While a recent review proposed targeting tissue-associated microbes for probiotic interventions[53], our denitrifying specialist isolates were predominantly obtained from the skeleton of *Oulastrea crispata*. This reveals the importance of often-overlooked skeleton-dwelling microbes in coral metabolism. Persistent intra-

population activity variations suggest further post-genomic regulation, warranting future multi-omics exploration.

While multivariate analysis linked denitrifying specialist enrichments in corals from western waters to both nitrate and phosphate gradients, functional validation through $^{15}$N-labeling assay confirmed nitrate as a critical driver of functional specialization. Notably, chronic nitrate pollution can destabilize coral-algal symbiosis by disrupting the nitrogen limitation necessary to control algal symbiont density and promote carbon translocation to the coral host[54]. Denitrifying specialists may help convert excess nitrate/nitrite to N$_2$, redirecting energy to coral host resilience in high-nitrate environments. Interestingly, specialists exhibit distinct genetic adaptations shaped by water nitrogen and phosphorus gradients, with metabolic pathways differentially present compared to non-specialists, i.e., a trend toward depletion of nitrogen transport, assimilatory nitrate/nitrite reductases, and phosphorus acquisition pathways. Specifically, the depletion of nitrogen transport systems (*nrtB/nasE/cynB*, *nrtA/nasF/cynA*, *nrtC/nasD*) reduces ATP expenditure on active transport, allowing for minimum nitrogen uptake in nutrient-rich waters[55]. Eliminating assimilatory *nitrate/nitrite* reductases (*nasC/nasA*, *nasE*) conserves energy, likely by prioritizing respiratory denitrification over biosynthetic nitrogen assimilation[56]. Downregulating phosphorus acquisition pathways (*LHPP*, *mdoB*, *phnG/H/I/L*) minimizes energy costs for phosphate scavenging by relying on ample ambient phosphate in waters[57]. Ancestral reconstruction analysis further consolidates the co-evolved genes tied to nitrogen and phosphorus metabolisms, reinforcing the role of water nutrient gradients in driving niche specialization.

Hong Kong's eutrophic western reefs represent critical "coral reef oases", i.e., marginal environments where corals persist despite extreme conditions[33]. Our study deciphers a putative mechanism underlying this resilience and resolves a central paradox: while dominant denitrifying genera like *Ruegeria* are ubiquitous across reefs, only specific corals thrive under chronic nutrient pollution. These denitrifying specialist populations efficiently convert excess nitrate to N$_2$ in our incubations, which may help maintain the nitrogen limitation required for a healthy coral-algal symbiosis in natural reefs. This finding demonstrates that coral adaptation to nitrate gradients could be mediated by association with specialized bacterial populations within a genus, a mechanism that would be masked by genus-level microbial profiling. Given the complexity of natural reef environments, the presence of denitrifying specialists is arguably one of the key factors contributing to coral survival in western Hong Kong waters under chronic nutrient pollution. Future work should employ laboratory experiments involving microbiome manipulation and nutrient gradients to validate the functional efficacy of denitrifying specialists on coral holobiont biology. While our study identifies population-level specialization within denitrifying *Ruegeria* as a putatively key mechanism in supporting coral resilience under high nutrients, the co-dominance of other denitrifying genera (e.g., *Pseudomonas*, *Jhaorihella*) suggests a potential for niche partitioning within the

holobiont's denitrification consortium, which warrants future investigation. As such, our research underscores that the functional capacity of a microbiome exerts itself at the sub-genus level, providing a compelling case for a population-resolved framework to understand host-microbe interactions and ecosystem functioning in a changing ocean.

## Methods

### Coral sampling and *Ruegeria* isolation

Coral species *Acropora solitaryensis* (Veron and Wallace, 1984), *Acropora cf. samoensis* (Brook, 1891), *Platgyra acuta* (Veron, 2000), *Favites adbita* (Ellis and Solander, 1786), and *Oulastrea crispata* (Lamarck, 1816) used for *Ruegeria* isolation were collected along nine western to eastern Hong Kong reef sites, i.e., Yam Tsai Wan (YTW), Sham Wan (SW), Lo Chau (LC), Bluff Island (BI), Ngo Mei Chau (NMC), Chek Chau (CC), Wong Wan Chau (WWC), Ninepine (NP), and Kiu Tsui Chau (KTC) between the years 2020 and 2022. Two mother colonies, each around 5 cm² in size, were collected for each coral species. Fresh coral samples were fragmented in situ with a hammer and chisel, stored in sterile Ziplock bags containing seawater, and transferred to the laboratory within 3 h. Following our previous study[44], corals were separated into three compartments, i.e., mucus, tissue, and skeleton, with each used for *Ruegeria* isolation, respectively. Specifically, we rinsed each coral colony three times using the 0.22 µm-filtered seawater (FSW) and inverted it for one minute to collect 1 ml of mucus. For corals (e.g., *Acropora* sp.) that did not produce enough mucus by inversion, we collected the mucus by pipetting against the single coral polyp. After rinsing corals with FSW three times to ensure no mucus was left, coral tissue was collected by being sprayed off from the skeleton using the Waterpik (USA) and FSW, centrifuged at 3000 × *g* and 4 °C for 10 min, and finally resuspended in 1 ml of FSW. The coral skeleton of approximately 5 cm² was ground into a coarse gravel size using the sterile mortar and pestle. They were resuspended in 1 ml of FSW and vortexed at 800 rpm for 3 min.

The collected bacterial suspensions were diluted for 10, 100, and 1000-fold using the FSW. 100 µl from each dilution was spread on the marine basal media (MBM) agar plates called MBM-Taurine, MBM-Dimethylsulfoniopropionate (DMSP), and MBM-Choline O sulfate (COS). All three media shared the same basic recipe, which contains K₂HPO₄, Tris-HCl, Sea Salt, Bacto Agar, FeNaEDTA, and Vitamin (see details in Supplementary Data 15). However, three media used different carbon and nitrogen sources, with MBM-Taurine using Taurine, glycine betaine, and N-acetylglucosamine, MBM-DMSP using DMSP and NH₄Cl, and MBM-COS using Choline-O-sulfate, glycine betaine, and N-acetylglucosamine, respectively. All three carbon and nitrogen sources were known as common osmotes from corals[58]. MBM agar plates were incubated at 28 °C for three weeks. *Ruegeria* colonies were picked according to the morphology and purified four times on the Difco™ Marine Agar 2216 (BD, USA). *Ruegeria* isolates were confirmed by 16S rRNA gene PCR and Sanger sequencing using the primer pair 16S-27F (5′-AGAGTTTGATCMTGGCTCAG-3′) and 1492R (5′-CGGTTACCTTGTTACGACTT-3′)[59]. The sequencing results were confirmed with BLASTn against the prokaryotic databases in EzBioCloud (accessed in January 2022). *Ruegeria* isolates were preserved in 75% glycerol at −80 °C, with their biomass collected for genomic DNA extraction.

### *Ruegeria* genomic library construction, sequencing, and sequence analysis

Genomic DNA of *Ruegeria* isolates was extracted with the TIANamp Genomic DNA Kit (TIANGEN Biotech, China) and quantified with the Qubit fluorometer (Invitrogen, USA) and dsDNA High Sensitivity Assay Kit (Invitrogen, USA). *Ruegeria* genome sequencing library was constructed using the NEBNext® Ultra™ II FS DNA Library Prep kit (NEB, USA). Specifically, we fragmented the genomic DNA to about 500 bp

and ligated the DNA for each *Ruegeria* genome with specific barcodes and adaptors. All *Ruegeria* genomes were pooled together for next-generation genomic sequencing on the platform DNBseq PE150 at BGI (BGI, China).

Raw next-generation sequencing (NGS) reads were quality-checked using FastQC (v0.11.8). Adapters and low-quality bases were removed using Trimmomatic (v0.39). Reads with a Phred score below 30 were discarded to ensure high-quality data. The remaining high-quality reads were assembled using the Shovill assembly pipeline, which employs SPAdes as its core with default parameters[60]. The completeness and contamination levels of the assembled genomes were assessed using CheckM (v1.1.3)[60], ensuring the minimum completeness of 50% and contamination below 10%. Protein-coding genes were predicted using Prokka (v1.14.6) with default settings for bacterial genomes[61]. Orthologous gene families were identified using Ortho-Finder (v2.5.1), incorporating 419 *Ruegeria* genomes along with 26 referenced *Ruegeria* genomes retrieved from public databases (Supplementary Data 4). Total 552 single-copy orthologous genes were aligned with MAFFT (v7.471)[62] and concatenated at the amino acid level for downstream phylogenetic analysis. A maximum likelihood phylogenomic tree was constructed using IQ-TREE (v1.6.12)[63] with the parameter: -s alignment -m MFP -wbtl -bb 1000 -mset WAG,LG,JTT,-Dayhoff -mrate E,I,G,I+G -mfreq FU. The embedded ModelFinder identified the best-fit model as JTT+I+G4 and ultrafast bootstrap analysis was conducted with 1000 replicates. Species delineation in 419 *Ruegeria* isolates was performed with PopCOGenT using default parameters, with the delineated populations referred to as main clusters (MCs)[39]. Functional annotation on each KO was performed with KofamScan using the latest KEGG database (updated on 2025/07/15). This version of the KEGG database includes comprehensive functional hierarchies for environmental energy metabolisms, such as carbon, nitrogen, phosphorus, and sulfur metabolisms[64]. It incorporated previously omitted pathways like Dimethylsulfoniopropionate (DMSP) degradation, thereby addressing limitations in annotating environmentally relevant pathways noted in earlier versions[65].

### Nanopore library construction, sequencing, and sequence analysis for *Ruegeria* closed genomes

To generate reference genomes for 19 phylogenetically diverse MCs, long-read nanopore sequencing was conducted on 34 representative isolates. MC10 (i.e., having high contig counts from NGS) and its sister group MC46 included 15 and two closed genomes, respectively. Each MC contained one representative closed genome for the remaining 17 MCs. DNA libraries were constructed using the Ligation Sequencing Kit (SQK-LSK110, Oxford Nanopore Technologies) and sequenced on a MinION device with an R9.4.1 flow cell. Genome assemblies were generated using three approaches: (i) Unicycler (v0.5.0) in hybrid mode with an overlap-layout-consensus (OLC) method[66], (ii) Flye (v2.6) employing a de Bruijn graph approach[67], and (iii) Canu (v2.2) using an OLC-based method[68]. Short-read NGS data were used to polish assemblies with Pilon (v1.24)[69]. Assembly quality was assessed with Mauve and Bandage, and circularized genomes were oriented using Circlator to position the *dnaA* gene at the origin. Protein-coding genes were annotated with Prokka (v1.14.6)[61] with genome completeness evaluated using CheckM (v1.0.7) and miComplete (v1.1.1)[70]. Functional annotations were conducted using KOfamscan, COG, and TIGRFAM databases with HMMER (v3.3) with an e-value cutoff of 1e-20[71].

### Denitrification gene arrangements and phylogenetic trees in *Ruegeria* genomes

The presence of denitrification genes (i.e., *napA/narG* for nitrate reductase, *nirK/nirS* for nitrite reductase, *norB* for nitric oxide, and *nosZ* for nitrous oxide reductase) in *Ruegeria* genomes (i.e., NGS sequencing-based) was screened with the KofamScan[72] using the KEGG database. The presence of denitrification genes in *Ruegeria* genomes

was visualized in the phylogenomic tree with the iTOL (v5)[73]. Phylogenetic trees for denitrification genes were generated by aligning each gene individually, concatenating the alignments, and inferring the tree using IQ-Tree with the auto mode selection mode. Denitrification gene arrangement analysis in *Ruegeria* genomes was performed using both NGS and long-read Nanopore sequencing data. For genomes assembled into single circular chromosomes (i.e., derived from Nanopore data), 30 kb regions upstream and downstream of each key gene were extracted and visualized using the pyGenomeViz. Different contigs were separated by double slashes ("//"). For genome pairs vertically adjacent in the phylogenomic tree, the denitrification genes and flanking regions were aligned using BLASTn with homologous segments indicated by gray shading.

## Primer design to resolve *Ruegeria* at the population level
To investigate *Ruegeria* community compositions in environmental samples, we designed two primer pairs that can resolve *Ruegeria* at the population level (Supplementary Fig. 9). For this, we constructed the phylogenetic trees for each of the single-copy orthologous genes in 419 *Ruegeria* genomes, with the other genera belonging to the Rhodobacteraceae family downloaded from NCBI as the outgroups. The phylogenetic trees showing the ability to resolve different *Ruegeria* MCs were kept for the primer design. Single-nucleotide polymorphism (SNP) sites in *Ruegeria* population-resolving gene sequences were used as the target regions for primer design. We designed the primers according to the following three criteria: (i) forward and reverse primers both contain at least one SNP site to differentiate *Ruegeria* from other genera; (ii) primer length is limited to 17–25 bp with the target amplicon size less than 600 bp to fit the next-generation sequencing platform; (iii) there is no complementarity between forward and reverse primers. Two primer pairs, i.e., 562 F (5'-GGYAAR-ACCGTTCTGATCATG-3') and 1111 R (5'-CYTTTTCCGARATCGCDCGG-3') for *ATP5B* (i.e., a gene encoding a subunit of mitochondrial ATP synthase), as well as 668 F (5'-CRCTGGTSGACGGGCARGG-3') and 1189 R (5'-CGATCAGYTTGGAYTTCTGRACCTG-3') for *parC* (i.e., a gene encoding DNA topoisomerase IV subunit A) were designed for *Ruegeria* population-resolving amplicons. The product sizes for *ATP5B* and *parC* amplicons were 550 bp and 522 bp, respectively. An additional *nirS* primer pair targeting denitrifying *Ruegeria* was designed following the same criteria mentioned above. The 382 bp of *nirS* PCR product was amplified by *nirS*-Rue514F (5'-GGCGGBATGAACAACTTYTCGG-3') and *nirS*-Rue895R (5'-GAYGGSAAGATGACYAARATCG-3').

## Coral sampling, amplicon sequencing, and sequence analysis
Five coral species, i.e., *Oulastrea crispata*, *Favites abdita*, *Leptastrea pruinosa* (Crossland, 1952), *Porites lutea* (Milne Edwards, 1860), and *Platygyra acuta* were collected from seven reef sites, i.e., YTW, PC, SW, LC, NP, Sharp Island (SI), and Port Island (PI) in November 2022 (Fig. 1a). These coral species were chosen in this study due to their consistent prevalence along western to eastern Hong Kong reefs. Fresh corals with a size of 5–10 cm$^2$ were collected with a chisel and hammer, placed in sterilized bags on ice, and immediately transported to the laboratory. Coral fragments were rinsed with FSW upon arrival and stored in a −80 °C freezer until further processing. DNA extraction for coral samples (i.e., mucus, tissue, and skeletal compartments) was performed with the DNeasy PowerSoil Pro Kits (Qiagen, Germany) according to manufacturer instructions. DNA yield was quantified using the Qubit fluorometer (Invitrogen, USA) and dsDNA High Sensitivity Assay Kit (Invitrogen, USA).

Amplicon libraries for *Ruegeria* population-resolving genes *ATP5B* and *parC* as well as *nirS* and *Ruegeria*-targeted *nirS* genes were constructed using the PCRs with identical dual barcodes and primers according to our previous study[74]. The *nirS* amplicon targets denitrifier communities across a large scale of taxonomy and was amplified using the primer pair *nirS*-1F (5'-CCTAYTGGCCGCCRCART-3') and *nirS*-qR (5'-TCCMAGCCRCCRTCRTGCAG-3'). The PCR condition for each target amplicon consisted of 5 µl Premix Taq polymerase (TaKaRa), 1 µl forward primer (5 µM), 1 µl reverse primer (5 µM), 2 µl H$_2$O, and 1 µl DNA template (-20 ng µl$^{-1}$). Thermal cycling conditions consisted of an initial activation at 95 °C for 3 min, followed by 35 cycles of 95 °C for 30 s, 55 °C (*ATP5B*) or 58 °C (*parC*) or 55 °C (*nirS*) or 52 °C (*Ruegeria*-target *nirS*) for 45 s, and 72 °C for 45 s, followed by a final extension at 72 °C for 10 min. PCR products were purified with MGIEasy DNA Clean Beads (MGI, China) and normalized with the SequalPrep Normalization Plate Kit (Invitrogen, USA). PCR products for each amplicon were pooled together for the adapter ligation using the MGI DNA Library Preparation Kit (MGI, China). A final DNA nanoball library was constructed for all pooled samples and sequenced on the DNBSEQ-G400RS PE300 platform at MGI (Hong Kong SAR, China).

Amplicon sequence analysis was performed in R (v3.5.1) following the DADA2 (v1.10.0) with slight modifications[75]. Primers and low-quality bases were removed from the demultiplexed sequences using Trimmomatic (v0.39)[76]. Reads were de-replicated, and error rates were estimated and used for inference of amplicon sequence variants (ASVs). ASV tables were constructed for four target gene amplicons (*ATP5B*, *parC*, *nirS*, *Ruegeria*-target *nirS*) with chimera removal using the de novo approach. For the taxonomic assignment of *nirS* amplicon-derived ASVs, we constructed the sequence alignment based on *nirS* sequences downloaded from the FunGene repository[77]. To assign ASVs from *ATP5B*, *parC*, and *Ruegeria*-target *nirS* amplicons to each *Ruegeria* MC, we aligned the ASV sequences to the corresponding gene sequences extracted from *Ruegeria* genomes using the reference-based alignment software Papara (v2.5) alignment[78]. For each amplicon, a gene tree was conducted using both amplicon-derived ASVs and *Ruegeria* genome-derived partial gene sequences at the nucleotide level with IQ-Tree (v1.6.12)[63]. Each ASV was assigned to a specific MC based on the parental node affiliation, which was determined if over 80% of the descending tips originated from a particular MC.

## Identification of denitrifying specialist *Ruegeria* in corals from high nitrate waters
Spatial distributions of *Ruegeria* MCs in corals along coastal Hong Kong reefs were shown based on ASV numbers from *ATP5B*, *parC*, and *nirS* amplicons using the non-metric multidimensional scaling (NMDS) ordination with Bray-Curtis distance. The function "envfit"[79] was applied to pinpoint the abiotic (i.e., environmental parameters) and biotic (i.e., *Ruegeria* MCs) variables that influenced *Ruegeria* population compositions across different coral species and sites. The length of the fitted arrows in the NMDS ordination represented the correlation efficiency ($R^2$) between driving factors and *Ruegeria* population compositions. In the Envfit analysis, $R^2$ (range: 0–1) represents the proportion of variance in amplicon data explained by the fitted (a) biotic vector (MCs or environmental factors), with $P$-values < 0.1 indicating weak, 0.1–0.3 indicating moderate, and >0.3 indicating strong correlations. The $P$-value below 0.05 denotes the statistical significance of this fit. *Ruegeria* MCs were classified as "denitrifying specialists" when they were enriched in corals from high nitrate western waters based on both *Ruegeria*-population resolving (*ATP5B* or *parC*) and denitrifying *Ruegeria*-population resolving *nirS* amplicons. In contrast, *Ruegeria* MCs possessing complete denitrification genes but failing to meet the above criteria were classified as "denitrifying non-specialists".

## Phylogenomic identification of functionally adaptive genes in denitrifying specialists
To investigate the convergent evolution of denitrifying specialist *Ruegeria* and their adaptations to seawater nutrient gradients, we performed comparative genomic analysis using Phylosig and binaryPGLMM[43]. A phylogenomic tree was constructed to represent the evolutionary relationships among 419 *Ruegeria* isolates. A KEGG

Orthology (KO) annotation table showing KO presence and absence across 419 genomes was compiled alongside a classification of each genome as either a "denitrifying specialist" or "denitrifying non-specialist". Nitrogen and phosphorus-related KOs were subset to test the hypothesis that convergent evolution of denitrifying specialists is linked to their adaptations to water nutrient gradients. For each KO, a phylogenetic signal was assessed using the "phylosig" function (method = "lambda") to compute the $P$-value. If the phylogenetic signal was significant ($P \leq 0.05$), a binary Phylogenetic Generalized Linear Mixed Model (binaryPGLMM) was applied to test for correlations between KO presence/absence and the specialist/non-specialist traits, yielding regression coefficients and $P$-values. If the phylogenetic signal was non-significant ($P > 0.05$), a chi-squared test was performed to evaluate associations, accompanied by the linear regression coefficients. KOs with NaN results (i.e., KOs showing the same pattern across all *Ruegeria* genomes) from the phylosig test were excluded from further analysis.

### Tracing functional adaptations of denitrifying specialists through ancestral reconstruction

To study gene gain and loss events in denitrifying specialist *Ruegeria*, we generated a representative dataset of 155 *Ruegeria* genomes, with one genome per *Ruegeria* MC. We trimmed a previously constructed species tree, which is used as the reference tree for ancestral reconstruction. OrthoFinder was used to perform orthologue inference on all protein sequences[80], resulting in the identification of 15,964 orthologue groups (OGs). To ensure robust ancestral reconstruction, we filtered OGs by retaining those present at least 50% of the 155 genomes, resulting in 3603 OGs for downstream analyses. For each OG, MAFFT (v7.471)[81] was used to build multiple sequence alignments at the amino acid level, followed by IQ-TREE (v1.6.12)[63] to construct a maximum likelihood phylogenetic tree with the best-fit substitution model automatically selected. The resulting bootstrapped gene trees were used as inputs for ancestral reconstruction in ecceTERA[82] and ANGST[83]. To pinpoint genes involved in the evolutionary adaptation of denitrifying specialists, we examined the reconciled gene trees and classified OGs as functionally adapted in denitrifying specialists if OGs from at least two different denitrifying specialists shared a common ancestor whose descendants included no gene from other MCs. Identified OGs were annotated using the KEGG database.

### Denitrification activity measurements using the stable isotope tracer

The $^{15}$Nitrogen ($^{15}$N) stable isotope labeling approach was used to determine the denitrification activity of *Ruegeria* isolates according to Yao et al. [50]. The protocol was initially validated in a preliminary experiment using six *Ruegeria* MCs representing genomically different denitrification phenotypes (including non-denitrifying *Ruegeria*, denitrifying specialists, and denitrifying non-specialists) under two different dissolved oxygen conditions (Supplementary Fig. 3). Second, we assessed the denitrification activity for four denitrifying specialist MCs (i.e., MC1, MC6, MC13, and MC18) and nine non-specialist MCs (i.e., MC0, MC3, MC7, MC12, MC17, MC19, MC20, MC21, and MC26) scattered and widely distributed in the genome tree. For this, three isolates were selected for each denitrifying specialist MC, and one to three isolates were selected for each denitrifying non-specialist MC based on isolate availability. To eliminate the possible interferences of ecological niches, we prioritized the isolate selection by covering different sampling sites (i.e., non-west vs. west) and coral compartments (i.e., mucus, tissue, skeleton) for both specialist and non-specialist MCs. Five biological replicates were used for each *Ruegeria* isolate and negative control group (i.e., no bacterial addition) in the $^{15}$N-labelling stable isotope assay. *Ruegeria* isolates were activated on the Difco™ Marine Agar 2216 (BD, USA) with their formed single colonies cultured in the Difco™ Marine Liquid 2216 (BD, USA). After 48 h of incubation to

reach the log phase, OD600 was measured for each *Ruegeria* isolate, with their colony-forming unit (CFU) counts correlated. We washed each *Ruegeria* isolate ($10^6$ CFU ml$^{-1}$) three times by a centrifugation at $3000 \times g$ for 5 min and resuspended it in the sterilized anaerobic flask with 20 ml of nutrient-depleting medium (Supplementary Data 16). This customized medium prevents the experiment from being affected by nitrate and nitrite availability in the Difco™ Marine Liquid 2216 medium. Given that denitrification usually occurs in the anaerobic environment[40], we created the nanaerobic condition (0.5 µmol l$^{-1}$ dissolved oxygen, DO)[41] by adding 99.8% pure argon gas into the flasks and monitored the DO concentrations in the liquid medium using the oxygen sensor (Pyroscience, Germany). After reaching the desired DO concentrations in the liquid medium, 0.1 mmol l$^{-1}$ of $^{15}$N-labeled sodium nitrate solution was added as the sole electron acceptor. After four hours of incubation at 28 °C and 150 rpm, 12 ml of headspace gas was collected and stored in the exetainer vials at 4 °C until further analysis.

The $^{15}$N-labeled $N_2$ and $N_2O$ concentrations were determined using purge and trap GasBench-Precon-Isotope-ratio mass spectrometry (IRMS) at the stable isotope facility of University of California, Davis. Stable isotope ratios of nitrogen ($\delta^{15}$N) are measured using a ThermoScientific GasBench + Precon gas concentration system interfaced to a ThermoScientific Delta V Plus isotope-ratio mass spectrometer (Bremen, Germany). Gas samples are purged from vials through a double-needle sampler into a helium carrier stream (20 ml min$^{-1}$). $N_2$ and $N_2O$ were isolated and concentrated in preparation for isotopic analysis. First, $N_2$ gas was sampled by a rotary 8-port valve fitted with a 5–100 µl sampling loop and timed to capture the peak $N_2$ concentration in the carrier gas stream. This gas sample was passed to the IRMS through an Agilent molecular sieve 5 A GC column (15 m × 0.53 mm ID × 50 µm film thickness, 40 °C, 2.2 ml min$^{-1}$). As $N_2$ was analyzed, the rest of the gas sample passed through a water trap ($P_2O_5$) and a $CO_2$ scrubber (Ascarite). $N_2O$ was trapped and concentrated in two liquid nitrogen cryo-traps operated in series, such that the $N_2O$ was held in the first trap until the non-condensing portion of the sample gas has been replaced by helium carrier before passing to the second, smaller trap. Finally, the second trap was warmed to ambient, and the $N_2O$ was carried by helium to the IRMS via an Agilent PoraPLOT Q GC column (25 m × 0.53 mm ID x 20 µm film thickness, 40 °C, 1.8 ml min$^{-1}$). This column separated $N_2O$ from residual $CO_2$. Calibration procedures for $N_2$ and $N_2O$ were applied identically across reference and sample materials and were directly traceable to the primary isotopic reference material for each element (i.e., Air for $\delta^{15}$N and VSMOW for $\delta^{18}$O). A pure $N_2$ or $N_2O$ reference gas was used to calculate provisional isotopic values of the sample peaks. Isotopic values were adjusted for changes in linearity and instrumental drift. Measurements were scale-normalized to the primary reference materials. The laboratory reference materials were mixtures of $N_2$ and $N_2O$. The $N_2$ was calibrated against an Oztech $N_2$ standard, but the calibration of the $N_2O$ was problematic since there were no suitable international standards. For $N_2O$, we calibrated $^{15}$N and $^{18}$O by thermally decomposing $N_2O$ in a heated gold tube (800 °C) to convert $N_2O$ to $N_2$ and $O_2$. The resulting $N_2$ was calibrated against the Oztech $N_2$ standard, and the $O_2$ was calibrated against an Oztech $O_2$ standard.

### Statistical analysis

Statistical analyses for denitrification activities measured by $^{15}$N-labeling stable isotope assay were conducted in R (v4.1.1) with several packages such as "ggplot2"[84], "lme4"[84], and "emmeans"[85]. To account for the phylogenetic non-independence of *Ruegeria* isolates and confounding effects of their evolutionary origins on denitrification activities (i.e., $^{46}N_2O$ and $^{30}N_2$), we performed the phylogenetic analysis of variance (PhylANOVA) using the "phylANOVA" function in the R package "phytools" (v2.4)[86]. The model included two fixed factors with

denitrification trait (i.e., specialists vs. non-specialists) as the primary predictor and MC identity as a covariate. The analysis was conducted with 10,000 simulations to generate a null distribution of the F-statistic under the Brownian motion evolution model. A phylogenomic tree for 29 *Ruegeria* isolates used in the $^{15}N$-labeling stable isotope assay was implemented in the PhylANOVA analysis. For this, single-copy ortho-logous genes across 29 *Ruegeria* genomes were aligned with MAFFT (v7.47)[81] and concatenated at the amino acid level. A maximum like-lihood phylogenomic tree was constructed using IQ-TREE (v1.6.12)[63] with the default parameters. To resolve denitrification activity varia-tions at the population and isolate levels, we fitted a linear mixed-effects model (LMM) defining the logarithm-transformed $^{46}N_2O$ and $^{30}N_2$ values as the responsible variables. The LMM included MC identity and isolate ID as two fixed effects with the biological replicate as a random effect. We graphically checked for all models whether the residuals were normally distributed using the histograms and quantile-quantile plots. The homoscedasticity of the residuals was checked by plotting them against the fitted values and each explanatory variable. Two-way Analysis of Variance (ANOVA) was conducted for the fixed effects using Tukey's Honest Significant Difference (HSD) as a post hoc. A chi-square test (i.e., Pearson's chi-square test of independence) was used to determine whether *Ruegeria* denitrification activity was significantly independent (no association) or dependent (associated) of isolate site (i.e., western vs. non-western reefs). For this, deni-trification activity was dichotomized as "group A" (higher than the median) or "group B" (lower than the median) in relative to the median $^{46}N_2O$ or $^{30}N_2$ production rates among the whole datasets. The total counts of western and non-western sourced isolates were statistically compared within each of the two groups for $^{46}N_2O$ or $^{30}N_2$ production rates, respectively.

### Reporting summary

Further information on research design is available in the Nature Portfolio Reporting Summary linked to this article.

## Data availability

The raw sequencing data generated in this study have been deposited in the NCBI database under several BioProjects. The raw reads derived from metabarcoding amplicons are available under NCBI BioProject accessions PRJNA1275576, PRJNA1275610, and PRJNA1310737. The raw reads of 409 *Ruegeria* isolates and genome assemblies of 419 *Ruegeria* isolates (10 genomes missing raw data) derived from next-generation whole genome sequencing are available under NCBI BioProject accession PRJNA1264799. The raw reads and genome assemblies of 34 *Ruegeria* isolates derived from Nanopore sequencing are available under NCBI BioProject accession PRJNA1275854. Source data are pro-vided with this paper.

## Code availability

Codes and scripts used for this study are available at: https://github.com/444thLiao/Denitrification_Ruegeria[87].

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

## Acknowledgements

We thank all the colleagues who have supported experiments on coral DNA extractions and $^{15}$N-stable isotope assay, including Kaitlyn E. Ho, Xinqing Lin, Johanna Fong, and Kai Yan Lai. We would like to thank Jinjin Tao and Xiaojun Wang for discussions on bioinformatics. Róisín Hayden, Emily Chei, Zhongyue Wan, Philip Thompson, and Joseph Brennan were acknowledged for assisting in coral sampling. H.L. discloses support for the research of this work from Hong Kong Research Grants Council (RGC) General Research Fund (project #: 14114724) and Marine Conservation Enhancement Fund (project #: MCEF21101). N.X., T.L., M.X., Z.Y.W., C.H.M., X.W.T., S.E.M., B.T., and C.R.V. declares no relevant funding.

## Author contributions

Conceptualization: H.L. Methodology and experimental investigation on coral sampling, *Ruegeria* isolation, and population-level primer design: M.X., T.L., S.E.M., and H.L. Methodology and experimental investigation on coral metabarcoding sequencing: N.X., T.L., and M.X. Bioinformatics: T.L. and N.X. Methodology and experimental investigation on $^{15}$N-stable isotope assay: N.X., Z.Y.W., C.H.M., X.W.T., and B.T. Statistical analysis: N.X. and T.L. Funding acquisition: H.L. Project administration: H.L. Supervision: H.L. Writing—original draft: N.X., H.L. Writing—review and editing: C.R.V., H.L., N.X., and S.E.M.

## Competing interests

The authors declare no competing interests.
