## [Transparent Peer Review file · Nature Communications]

Decoding Coral Resistance to Eutrophication through the Association of Hyper-Efficient Denitrifiers as Key Microbial Allies

Corresponding Author: Dr Haiwei Luo

Version 0:

Reviewer comments:

Reviewer #1

(Remarks to the Author)

Xiang and team bring together culturing, metabarcoding, population genomics, and activity measurements to disentangle differences in populations and activity of denitrifiers in corals subjected to high and moderate nutrient pollution. The study tackles an important knowledge gap, is well designed, and draws important conclusions. My expertise lies in genomics and biogeochemistry, and while I enjoyed reading this manuscript I found some parts difficult to follow, especially the many phylogenetic inferences used to disentangle relationships among the denitrification genes.

The work provides new and sound results showing that key microbial associates can cycle nitrite efficiently and possibly provide benefit to the coral host. The work is original and provides evidence that subspecies (strains) populations can be responsible for key aspects of the coral holobiont physiology. My understanding is that denitrifiers specialists were isolated from the skeleton and almost entirely from one coral species. Please highlight this better throughout the text. In general, the results support the conclusions partially, it is, in my opinion, not possible to infer whether the success of coral exposed to higher nutrient loads depends exclusively by the studied nitrifier populations. To conclusively demonstrate this, it would require laboratory experiments involving nutrient gradients and microbiome manipulation. The data analysis and interpretation are correct but the manuscript needs to be revised toning down some statements and clearly highlighting that the investigate denitrifier populations alone may not be at the base of the success of Hong Kong high nutrient corals.

It's unclear why the abstract often refers to denitrifying genera, stating for instance "our work resolves the paradox of why dominant denitrifying genera are ubiquitous", but then the metabarcoding analysis reveals that subspecies populations are actually the important denitrifiers.

The introduction is well written and effectively highlights existing knowledge gaps. However, it does not clearly explain how nitrate stress impairs key aspects of coral biology. Specifically, the consequences for growth, reproduction, and calcification in coral populations inhabiting poor water quality reefs are not addressed.

Materials and Methods section: be consistent in reporting packages version inside or outside brackets.

Line 59: 'another insidious threat namely nutrient pollution, is increasingly' is repeated twice

Line 73-74: unclear sentence.

Line 91: Obstructing?

Line 92: eutrophication or nutrient pollution? The two are not synonymous

Line 128: *Ruegeria* does not seem the dominant from Fig 1b, *Jhaorihella* seems equally abundant.

Line 175: It is unclear what are the colours in the rings around the phylogenetic trees of Figure 2C.

Line 179: in Fig 2D, I am unsure why there are arrows (explanatory variables) in these plots.

Line 205: How can a denitrification phenotype can be defined as non-denitrifying?

Line 233-234: I have some issues with this sentence. It is clear from Figure 3A that denitrifiers specialists are skeleton dwellers and come from *Oulastrea crispata*. Please rectify or better explain this sentence. I think some nuances of the data analysis should be better explained throughout the text to have a more transparent manuscript.

Line 247-248: This sentence should be modified 'suggesting convergent evolution possibly through horizontal gene transfer (HGT)'. HGT and convergent evolution are distinct mechanisms, and combining them in the same explanation is confusing and biologically inaccurate in this context.

Line 281-284: This statement does not make sense to me.

Line 285: marginally significant? Isn't $p=0.092$ non-significant?

Line 303: again convergent evolution and HGT are different things.

Line 316- 320: this is a long shot, please tone down.

Line 380-395: this paragraph is extremely speculative and not based on the evidence provided by the data.

Fig 1C: fix 'coral compartment'

Fig 2F: add x-axis label

(Remarks on code availability)
The results seem reproducible.

Reviewer #2

(Remarks to the Author)

Xiang et al present a comparative genomic analysis of coral-associated *Ruegeria* found across Hong Kong's coral reefs. Coral reefs in Hong Kong's western regions are subject to high nitrogen concentrations compared to reefs in eastern Hong Kong. Xiang et al show that *Ruegeria* populations differ across this geography, wherein isolates from western areas show markedly higher denitrification activity compared to those from central/eastern areas despite most isolates possessing the genetic machinery for denitrification. The manuscript focuses on *Ruegeria* because "The predominance of *Ruegeria* further confirms its suggested important role in coral-associated denitrification process" (line 127). The relative abundances quoted at line 126 appear to be averaged across all reefs, and Figure 1B indicates that other denitrifying genera such as *Pseudomonas*, *Jhaorihella* and *Bradyrhizobium* are more relatively abundant than *Ruegeria* at YTW site which has the highest concentration of nitrogen. I think it would be interesting to address the extent *Ruegeria* contributes to the adaptation/survival of its coral hosts in these high N environments.

One recommendation would be to relate the MCs to species and/or strain thresholds eg average nucleotide identity or relative evolutionary distance. Is an MC comparable to a *Ruegeria* species or strain, and do these MCs represent novel or existing *Ruegeria* species? If I understand correctly, PopCOGenT is based on gene flows which could become complicated depending on gene transfer events and geographic isolation? The genomes are mostly referred to as populations throughout the manuscript but also as strains in places (eg line 465).

More generally, I feel that it would improve readability of the manuscript if authors can orient readers by briefly explaining aims and methodology before describing results. For example, this paragraph to the next (line 177) goes from population genomics to amplicon profiling which won't be immediately clear to non-specialist readers. I understand this is largely personal preference in writing styles, but would be great to see explanation of the rationale and methods such that continuity between sections is more obvious.

Comments

Line 94: are the nitrate/nitrite/ammonia concentrations consistently high in western HK throughout the year, or do they spike during certain seasons e.g. increased runoff and discharge in wet seasons? Vice versa for eastern waters.

Fig. 1C: This information is in the figure legend but should be included in the methods too- the genome tree in fig 1C has 26 reference *Ruegeria* genomes which should be labelled to provide context to the tree. Indicate what species and accessions to identify them, do they fall within any of the MCs, i.e. do the MCs represent known *Ruegeria* species.

Line 165: It is not clear what data was used to construct the phylogenetic trees in figure 2C. On first read I thought it was based on ATP5B, parC and nirS genes from the isolate genomes to demonstrate that the three genes have sufficient resolution to distinguish between the 29 main clusters. The figure legend, however, says that trees are "based on

concatenated single-copy orthologous genes". At line 173, it is stated "phylogenetic trees constructed from amplified regions of all three genes...". Can authors clarify what these trees are based on and generally edit this section to make it clear what was done.

Line 186: I think it would help readability if authors add slightly more description here as the current text suggests that envfit identified MCs in amplicon data. Maybe a brief line before this to indicate that the ASVs were assigned to MCs identified from the genome tree by aligning ASVs to ... , followed by "envfit on the amplicon profiles showed that several MCs were enriched in high-nitrate waters."

Line 194: This statement cannot be verified as MC7 and MC40 are not shown in Fig. 2F. Maybe include a plot showing breakdown of the community by all MCs in addition to the four specific MCs in fig 2F. It would be interesting to know which MCs are more abundant in the low nitrate eastern waters.

Line 226: "thus the strains affiliated with specialist MCs were not necessarily isolated from the western sites only". Correct me if I am mistaken- since MCs are derived from phylogenetic clusters on the genome tree (figure 1C), it is possible that within each MC their nirS sequence (or other genes) can differ- this is shown in fig 2C where the MCs are not entirely monophyletic in the gene trees. As such could there be cases where isolates from for example MC6 may not necessarily be a "specialist denitrifier" as their copy of denitrification genes could be a "low efficiency" version?

Related to above: how do these MCs relate to conventional species thresholds- are the MC members "strains" (i.e. same species) as authors refer to in the manuscript, or are they different species of *Ruegeria*? Maybe check ANI or GTDB relative evolutionary divergence thresholds.

Line 340: "Denitrifying specialist strains MC1-AF0 and MC6-O11 achieved exceptional N₂ production rates (~480 and 210 pmol N₂ ml⁻¹ h⁻¹, respectively), surpassing a known denitrifying *Methylomirabilis* bacterium by three orders of magnitudes under a comparable 15N-labeling condition (estimated from 1.5 μmol 30N₂ productions by 50 mg wet biomass (~1010 CFU) in 150 ml anoxic serum vial over four hours)." I feel that this comparison could come across as disingenuous since authors claim that large phenotypic variability exists within *Ruegeria* populations, and as such the same logic should also apply to *Methylomirabilis* (i.e. does this specific *Methylomirabilis* species/strain have the highest denitrifying activity among other *Methylomirabilis*), and why compare *Ruegeria* with this *Methylomirabilis* isolated from paddy soils? It would be more relevant if authors compared with other denitrifiers in corals instead, eg *Pseudomonas*, *Jhaorihella* which were more relatively abundant than *Ruegeria* in YTW.

Minor comments

Line 49: superior efficiency compared to?

I suggest removing filler text eg "vibrant bastions of marine diversity" (line 56), "insidious" (line 59)

Line 63: it wasn't immediately clear what algae is referred to since nitrogen pollution also stimulates macroalgal proliferation. Maybe *Symbiodinium* algae?

Line 91: perhaps a typo – eighty coral species "obstructing" the reefs?

Line 102: not clear whether "fine-scale population (i.e. sub-species level)" refers to coral or microbial populations.

Figure 1A: typo "Ammnoia"

Figure 1C: typo "Coral ompartment". Maybe also indicate in the legend criteria for defining main clusters, e.g. 95% ANI?

Line 150: I would suggest changing "detoxifying eutrophic reefs"- detoxifying does not sound appropriate in this context.

Line 516: what is "mb"?

Line 183: This sentence needs editing "This stark contrast between the site-specific functional community structure and the stable genus-level abundance, underscoring the critical population dynamics we aimed to uncover."

Line 194: Fig. 2F is referenced in text before 2E, suggest swapping panels.

Line 276, 285, 288, 296: typo "BinayPGLMM"

Line 424- I believe should be "recipe" not "receipt"

Line 478: Not sure what is meant by "17 MCs included per closed genome per MC"

Line 514: "The phylogenetic trees showing the ability to resolve different *Ruegeria* MCs were kept for the primer design". Was there a measure of congruence between the gene (fig2C) and genome tree (Fig 1C)? I believe some of the MCs in fig2C are polyphyletic. Maybe label the MCs in fig2C as some of the colours are difficult to tell apart, and I think some MC colours do not appear in the tree in Fig1C? Do the colours in Fig1C and all panels in fig2C represent the same MCs?

(Remarks on code availability)

Reviewer #3

(Remarks to the Author)

In this manuscript Xiang et al. examine the role of denitrifying microbial partners within the coral microbiome and their contribution to resilience in the face of increasing eutrophication along Hong Kong's coral reef. The study sets out to understand why some reefs in high concentration nitrate/nitrite areas exhibit resilience and resist dysbiosis, which would otherwise be expected from the excess nitrogen leading to the proliferation of Symbiodiniaceae. The authors conclude that it is due to the presence of dominant denitrifying genera like *Ruegeria*. *Ruegeria*, through genomic inquiry, is demonstrated to have the genetic capability to fully reduce nitrate and nitrite down to N₂, which is not bioavailable for assimilation. Furthermore, although *Ruegeria* is ubiquitous across the nitrogen gradient of the Hong Kong Bay, it is demonstrated that specific strains are much more adept at this reduction and are able to execute it with upwards of 10-fold efficiency than others. There is a selective association of these efficient reducers with coral colonies in high nitrogen areas and the authors put forward that it is this metabolic process that curbs dysbiosis by limiting bioavailable nitrogen sources to the coral colony and microbiome.

Line 60: The clause "another insidious threat namely nutrient pollution, is increasingly" is repeated.

Line 84: Is it known which sections of denitrification gene clusters need to be intact within a genome for functional expression of that process?

Line 106: Have the authors investigated the dispersion of the ubiquitous bacteria in the water column across the area comprising the nitrate gradient? If there is equal dispersion, it would point more towards coral animal selectivity of denitrifying bacteria. If denitrifying bacteria are more concentrated in areas where they can make a better living, however, them ending up within the microbiomes of the corals there may be a byproduct of free association.

Line 121: "The *nirS* amplicon analysis revealed that denitrifier abundance in corals were evenly distributed at the genus level along water nitrate gradients." To my last point, this was determined through analysis of their presence in the coral mucus, tissue, and skeleton, correct?

Line 127: "The predominance of *Ruegeria* further confirms..." Does the relative stability of *Ruegeria*'s proportionality within populations across the nitrate gradient complicate the author's point that it aids resilience for corals at the high end of the gradient? How does this correlate to nitrate-driven selection?

Line 145: I see that last point gets picked up here!

Line 155: This is an apt explanation of what was done although, as written, it might be more appropriate in the methods section.

Line 209: "Denitrifying specialist *Ruegeria* MCs exhibited significantly higher productions..." belies statistical comparisons that were conducted between production levels. Those values should be included. Also, the text says that the denitrifying specialists were compared to the non-specialists, which suggests a comparison of two groups. The previous sentence states that four specialist MCs and nine non-specialist MCs (13 total) were being considered. It is slightly confusing here if the MCs are considered independently or as a group. In that case that they are considered as a group, can the authors comment on their preference for an ANOVA, which is typically used for comparing three or more groups. If, however, more than two MCs are being compared independently across phylogenetic lineage, that should be made clearer in the text.

Line 210: ⁴⁶N₂O would more accurately be notated as ¹⁵N₂O to denote that it is the stable isotope version of N₂O being investigated for the calculation. Similarly, for ³⁰N₂

Line 253: Extended Data Figs 4 & 5, numerous labels overlap making the annotations illegible. Consider offsetting some of the labels.

Line 348: "Quantification of end-product...specialist MCs." This is alluding to non-specialist MCs lacking the ability to fully reduce to N₂, correct?

Line 384: Can the authors comment on if the "selective enrichment" within a dominant genus like *Ruegeria* is host derived? Furthermore, all things being equal, what might prevent a colony within a eutrophic area with a specialist MC population in the water column from being colonized by it? Why aren't the benefits of this advantageous population ubiquitous throughout the zone?

Line 646: This is a strange way to end the sentence with reference to a publication. Perhaps you want to write, "...according to Yao et al.⁵⁰"

Line 679: MS data should be put into a repository, such as MassIVE, for the community.

Line 813: It is not immediately clear what statistical comparisons are being made in panels B & C. I recommend removing the legend text referencing isolate site and MC from the main plot field of panel B, that would then leave the field open to clearly denote what comparisons from the applicable results section are being referenced to in the plots.

General note on the figure scale, I assume this will be adjusted when the manuscript goes to print? They are quite large and not formatted for a printed page, which makes them difficult to consume and reference in the course of reviewing, although they are generally visually appealing.

(Remarks on code availability)

I think the README is sufficient for the genomic data.

Version 1:

Reviewer comments:

Reviewer #1

(Remarks to the Author)

I commend the authors for the improvement to the manuscript. I have a few more comments that I would like to see addressed before publication and that I think would improve the reach of this study as well as its relevance to the coral microbiology literature.

In the discussion I suggest that the authors articulate the involvement of skeleton-dwelling microbes in the coral holobiont metabolism. Often, skeleton-dwellers are not considered important to the holobiont physiology, for instance, this recent piece (<https://doi.org/10.3390/microorganisms14010202>) suggest that tissue associated microbes should be targeted for future probiotic strategies. Instead, the present study proves that skeleton-dwellers have important roles in the overall holobiont physiology and this should be articulated.

Line numbers referring to the clean version of the manuscript.

53-54: is this sentence complete? It reads awkwardly.

138-139: I suggest changing the title to "A majority of coral-associated *Ruegeria* isolates carry a complete denitrification gene set"

155-156: what was the completeness of the genomes that possessed a partial denitrification gene set or lacked denitrification genes? Could it be that they didn't have these genes because of sequencing bias? I think this point should be made explicit in the text.

(Remarks on code availability)

The README file is fine

Reviewer #2

(Remarks to the Author)

I thank the authors for addressing all reviewer comments, and for explaining the concept of main clusters (MCs) used in this manuscript. One final comment I have is regarding the use of the term "sub-species groups" when referring to the MCs. As authors pointed out that since MCs are distinct from the conventional species or strain level taxonomic ranks, I wonder if it still makes sense to refer to the MCs as sub-species populations (e.g. at lines 280, 299, 892, and 901 of the revised manuscript)? I would think "... driven by the association with specific, hyper-efficient denitrifying populations within the dominant genus *Ruegeria*" (line 892) or "capacity for associations of microbial allies at a MC population level" (line 901) would be more accurate in these contexts.

(Remarks on code availability)

Reviewer #3

(Remarks to the Author)

I have considered the authors review letter with their responses to my and my colleagues concerns from the initial review and the updated manuscript. I find my concerns to have been adequately satisfied.

I would implore the authors to carefully review the figures. Panel 1A has location markers that obscure the location text. Panels 3N and 3C have ill spaced facets that are not centered. Prior to publication these noticeable details should be fixed to present a polished final product to the community.

Beyond that, I have no further suggestions at this time.

(Remarks on code availability)

Decoding Coral Resistance to Eutrophication: Association of Hyper-Efficient Denitrifiers as Key Microbial Allies

We would like to thank the Nature Communications editors for considering our manuscript. We have provided detailed point-by-point responses to each of the reviewer's points in **blue** and highlighted changes to the manuscript in **green** (which are also pointed out in the track-changes version of the manuscript file). Please note that the line numbers cited in our responses refer to the revised version with track changes.

REVIEWER COMMENTS

Reviewer #1 (Remarks to the Author):

Xiang and team bring together culturing, metabarcoding, population genomics, and activity measurements to disentangle differences in populations and activity of denitrifiers in corals subjected to high and moderate nutrient pollution. The study tackles an important knowledge gap, is well designed, and draws important conclusions. My expertise lies in genomics and biogeochemistry, and while I enjoyed reading this manuscript I found some parts difficult to follow, especially the many phylogenetic inferences used to disentangle relationships among the denitrification genes.

Response: We thank the reviewer for the positive evaluation and for raising this point, which allows us to clarify the purpose and interpretation of the phylogenetic analyses presented in the study. Below, we summarize the distinct roles of the phylogenies shown throughout the figures with relevant text sections highlighted in the revised manuscript (lines 239-242, 386-390, and 508-513).

1. Fig. 1c - Phylogenomic tree of all *Ruegeria* isolates with denitrification gene profiles. This tree is based on a core-genome alignment of the 419 newly sequenced *Ruegeria* isolates and 26 publicly available reference genomes. Specifically, a set of orthologous genes shared by all 445 analyzed *Ruegeria* genomes were identified with their amino acid sequences aligned and concatenated to identify similarities and differences at each position. The pattern of variations in this core-genome alignment is used to reconstruct an evolutionary genome tree. Isolates with more similar core-genome sequences are placed on closer branches, indicating they share a more recent common ancestor. The ecological niche information and distribution of key denitrification genes were visualized in the phylogenomic tree to demonstrate that over 80% of *Ruegeria* isolates possess the genetic potential for complete denitrification.

“A phylogenomic tree of *Ruegeria* isolates, constructed using a core-genome alignment of these isolates, along with 26 publically available *Ruegeria* genomes as the reference (Fig. 1c; Supplementary Dataset 3), was used to depict the distribution of key denitrification genes across isolates within this genus.”

2. Fig. 3a - Phylogenomic tree of the 29 representative *Ruegeria* isolates used in the ¹⁵N-stable isotope assay. This tree is constructed based on a core-genome alignment of 29 *Ruegeria* isolates assayed in the ¹⁵N-stable isotope assay. This tree was used as the input for phylANOVA (Phylogenetic ANOVA) analysis, allowing comparisons of denitrification activity (e.g., specialist vs. non-specialist MCs) while accounting for the shared evolutionary history. This phylANOVA statistical method controls for phylogenetic non-independence of 29 *Ruegeria* isolates and strengthens the statistical validity of denitrification activity trait-based comparisons.

“The Phylogenomic tree of the 29 assayed *Ruegeria* isolates was used for the phylANOVA (Phylogenetic ANOVA) analysis, which controls for an evolutionary dependence of isolates during the comparisons of denitrification activity between specialist vs. non-specialist MCs.”

3. Extended Data Fig. S3 (now is Supplementary Fig. 5) - Gene trees of key denitrification operons of all *Ruegeria* isolates. Individual maximum-likelihood trees were constructed for the core genes *napA*, *nirS*, *norB*, and *nosZ* using partial target sequences from the isolates. These trees confirm that phylogenetic patterns were consistent with convergent evolution of denitrifying specialist MCs, independent of core-genome ancestry.

“Denitrification genes from denitrifying specialists and non-specialists may have different evolutionary histories. Individual maximum-likelihood gene trees of key denitrification operons (*nap*, *nir*, *nor*, *nos*) were constructed at the amino acid level using the extracted gene sequences from *Ruegeria* genomes. Comparison of these phylogenetic trees and the phylogenomic tree revealed incongruent evolutionary patterns for denitrification genes in denitrifying specialists.”

We hope these revisions clarify the phylogenetic methods and analytical rationales.

The work provides new and sound results showing that key microbial associates can cycle nitrite efficiently and possibly provide benefit to the coral host. The work is original and provides evidence that subspecies (strains) populations can be responsible for key aspects of the coral holobiont physiology. My understanding is that denitrifiers specialists were isolated from the skeleton and almost entirely from one coral species. Please highlight this better throughout the text. In general, the results support the conclusions partially, it is, in my opinion, not possible to infer whether the success of coral exposed to higher nutrient loads depends exclusively by the studied nitrifier populations. To conclusively demonstrate this, it would require laboratory experiments involving nutrient gradients and microbiome manipulation. The data analysis and interpretation are correct but the manuscript needs to be revised toning down some statements and clearly highlighting that the investigate denitrifier populations alone may not be at the base of the success of Hong Kong high nutrient corals.

Response: We thank the reviewer for their acknowledgement of the conceptual advance of our manuscript and raising these important points. Regarding the isolation source, we confirmed that all available isolates belonging to the denitrifying specialists were isolated from *Oulastrea crispata* (predominantly from the skeleton, with one exception from the mucus). We have highlighted this finding in lines 359-361 in the revised manuscript.

“Notably, available isolates of denitrifying specialists were all sourced from the coral species *Oulastrea crispata*, with many of them being isolated from the skeleton except one, which was obtained from the mucus layer.”

Regarding the data interpretation for ecological implications, we have toned down the statements in the Results and Discussion sections in the revised manuscript. For example, we have highlighted “Given the complexity of natural reef environments, the presence of denitrifying specialists is arguably one of the key factors contributing to coral survival in western Hong Kong waters under chronic nutrient pollution.” in lines 993-996 in the revised manuscript.

We agree that controlled microbiome manipulation experiments with nutrient gradients represent a critical next step to validate the putative role of denitrifying specialists in coral holobiont fitness. While such complex mesocosm manipulations are beyond the scope of the present study, we now highlight this as an important direction for future research in the lines 996-998 in the revised Discussion part.

“Future work should employ laboratory experiments involving microbiome manipulation and nutrient gradients to directly validate the functional efficacy of denitrifying specialists on coral holobiont biology.”

It's unclear why the abstract often refers to denitrifying genera, stating for instance “our work resolves the paradox of why dominant denitrifying genera are ubiquitous”, but then the metabarcoding analysis reveals that subspecies populations are actually the important denitrifiers.

Response: Apologies if this was unclear. In the original sentence, we aim to highlight that merely looking at denitrifying genera across the nitrate gradient would mask critically important, adaptive differentiation occurring at the sub-species population level. This sentence meant to emphasize that our research resolves this precise paradox: although dominant denitrifying genera can be ubiquitous, corals can exhibit marked differential resilience based on fine-scale, population-level specialization within these genera, not accessible by conventional metabarcoding approaches. We believe this original sentence delivers our central message: critical microbially mediated ecological adaptation can be hidden at the genus level, which requires sub-species population-level investigations. Please see the full sentence in lines 52-55 in the revised manuscript, as detailed below.

“As such, our work resolves the paradox of why dominant denitrifying genera are ubiquitous, yet only certain corals thrive in eutrophic conditions, while providing a framework for future studies delineating ecologically important host microbes.”

The introduction is well written and effectively highlights existing knowledge gaps. However, it does not clearly explain how nitrate stress impairs key aspects of coral biology. Specifically, the consequences for growth, reproduction, and calcification in coral populations inhabiting poor water quality reefs are not addressed.

Response: We thank the reviewer for the advice. We have added some relevant text to expand the detrimental effects of excess nitrate stress on coral growth, reproduction, and calcification in lines 68-70 in the revised Introduction part.

“Nitrate enrichment induces holobiont oxidative stress that restricts energy resources for coral growth and reproduction ⁹, and interferes with biomineralization, leading to reduced calcification rates ⁵.”

Materials and Methods section: be consistent in reporting packages version inside or outside brackets.

Response: Thank you. We have reported the package versions inside brackets throughout the manuscript.

Line 59: ‘another insidious threat namely nutrient pollution, is increasingly’ is repeated twice

Response: We have deleted the repeated sentence in the revised manuscript.

Line 73-74: unclear sentence.

Response: To clarify this point, we have added “coral autotrophic capability” in line 80 in the revised manuscript.

Line 91: Obstructing?

Response: We apologized for the typo, which has been updated as “inhabiting” in line 97 in the revised manuscript.

Line 92: eutrophication or nutrient pollution? The two are not synonymous

Response: Thanks for pointing this out. We have corrected it as “nutrient pollution” in lines 98-99 in the revised manuscript.

Line 128: *Ruegeria* does not seem the dominant from Fig 1b, *Jhaorihella* seems equally abundant.

Response: We thank the reviewer for this careful observation. *Ruegeria* and *Jhaorihella* appear visually comparable in the Fig. 1b, yet the quantitative community analysis showed that *Ruegeria* (~12%), *Pseudomonas* (~12%), *Jhaorihella* (~10%), and *Halomonas* (~7%) were the top four dominant genera within coral-associated *nirS* denitrifying bacterial communities across coral species and sites. Following this, we have replaced the word “dominance” with “high abundance” in lines 225-229 in the revised manuscript.

“At the genus level, denitrifier communities in Hong Kong reef-building corals were largely similar across coral species and sampling sites at the genus level, with *Ruegeria* (~12%), *Pseudomonas* (~12%), *Jhaorihella* (~10%), and *Halomonas* (~7%) dominating (Fig. 1b). The high abundance of *Ruegeria* further confirms its suggested importance in coral-associated denitrification.”

Line 175: It is unclear what are the colours in the rings around the phylogenetic trees of Figure 2C.

Response: We thank the reviewer for raising this point. The color annotation “*Ruegeria* populations (i.e., main clusters, MCs) delineated with PopCOGenT were shown in different colors in the trees.” was previously provided in the legend text of Fig. 2c. Phylogenetic trees constructed from amplified regions of all three genes confirmed their utility by grouping *Ruegeria* MCs into distinct clusters, validating their power to resolve different populations. To improve clarity, we have added the color annotation in the revised Fig. 2c.

Line 179: in Fig 2D, I am unsure why there are arrows (explanatory variables) in these plots.

Response: Thank you for raising this point. The arrows in the NMDS plots (Fig. 2d) are explanatory vectors resulting from the envfit analysis, a standard method in community ecology (Oksanen et al., 2001). The arrows pointed to the biotic driving forces (i.e., populations) for *Ruegeria* community compositions across coral species and sampling sites. The meaning of arrow direction and length is explained below.

Direction: The direction of each arrow indicates the gradient of increase for a particular variable (i.e., MC identity in Fig. 2d) across the ordination plot. Samples projected in the direction of the arrow have high values for that variable.

Length: The length of the arrow is proportional to the correlation strength between the variable and the *Ruegeria* community composition. A longer arrow means that variable is a stronger driver of the observed community distribution.

Envfit analysis visually confirms our key finding stated in the text: specific *Ruegeria* MCs were significantly associated with corals from high-nitrate waters. For example, the long arrow pointing toward specialists (i.e., MC1, MC6, MC13, and MC18) directly illustrates their enrichments in high-nitrate western waters. We have clarified this in the revised figure legend. Please refer to lines 1577-1580 in the revised manuscript.

“Arrow direction indicates the gradient of increase for a particular variable across the ordination plot, with samples projected in the arrow direction owning high values. Arrow length indicates the correlation strength between a particular variable and *Ruegeria* community composition.”

Oksanen, J., Simpson, G. L., Blanchet, F. G., Kindt, R., Legendre, P., Minchin, P. R., ... & Weedon, J. (2001). *Vegan: community ecology package*.

Line 205: How can a denitrification phenotype can be defined as non-denitrifying?

Response: We agree that “non-denitrifying” is vague. We have replaced it by “a negative control lacking denitrification genes” in line 383 in the revised manuscript.

Line 233-234: I have some issues with this sentence. It is clear from Figure 3A that denitrifiers specialists are skeleton dwellers and come from *Oulastrea crispata*. Please rectify or better explain this sentence. I think some nuances of the data analysis

should be better explained throughout the text to have a more transparent manuscript.

Response: Yes the reviewer is correct. Our available isolates of denitrifying specialists were all sourced from *Oulastrea crispata*, with most of them being isolated from the skeleton excepting one obtained from the mucus. We have revised the phrasing as in lines 359-361.

“Notably, available isolates of denitrifying specialists were all sourced from the coral species *Oulastrea crispata*, with many of them being isolated from the skeleton except one, which was obtained from the mucus layer.”

Regarding statistical analyses, we have provided rationales for them in lines 386-390, 494-500 in the revised manuscript.

“The phylogenomic tree of the 29 assayed *Ruegeria* isolates was used for the phylANOVA (Phylogenetic ANOVA) analysis, which controls for an evolutionary dependence of isolates during the comparisons of denitrification activity between specialist vs. non-specialist MCs.”

“To test the correlation between the geographical origin of isolates (western vs. non-western sites) and their denitrification activity phenotypes, we performed chi-square tests, which are suitable for analysing association between categorical variables.”

Line 247-248: This sentence should be modified ‘suggesting convergent evolution possibly through horizontal gene transfer (HGT)’. HGT and convergent evolution are distinct mechanisms, and combining them in the same explanation is confusing and biologically inaccurate in this context.

Response: HGT is an important mechanism leading to convergence (Lozupone et al., 2008; Arnold et al., 2022). The spread of resistance genes across diverse bacteria via mobile elements (plasmids, transposons) is a classic example, leading to convergent resistance phenotypes (Castañeda-Barba et al., 2024). However, many other mechanisms can also lead to convergent evolution in bacteria, such as convergent gene loss, convergent amino acid replacements, and convergent evolution at higher order (Cao et al., 2025). As such, we respectfully think the original sentence is appropriate, which indicates that HGT is one of the mechanisms for convergent evolution.

Lozupone, C. A., Hamady, M., Cantarel, B. L., Coutinho, P. M., Henrissat, B., Gordon, J. I., & Knight, R. (2008). The convergence of carbohydrate active gene repertoires in human gut microbes. *Proceedings of the National Academy of Sciences*, 105(39), 15076-15081.

Arnold, B. J., Huang, I. T., & Hanage, W. P. (2022). Horizontal gene transfer and adaptive evolution in bacteria. *Nature Reviews Microbiology*, 20(4), 206-218.

Castañeda-Barba, S., Top, E. M., & Stalder, T. (2024). Plasmids, a molecular cornerstone of antimicrobial resistance in the One Health era. *Nature Reviews Microbiology*, 22(1), 18-32.

Cao, Z., Zhang, H., & Zou, Z. (2025). Language models reveal a complex sequence basis for adaptive convergent evolution of protein functions. *Proceedings of the National Academy of Sciences*, 122(39), e2418254122.

Line 281-284: This statement does not make sense to me.

Response: We agreed that the direct link between “pathway depletion” and “energy conservation” was speculative. In the revised manuscript, we have restructured this section to present a more nuanced and logically coherent argument. Please refer to in lines 705-709.

“The coordinated decreasing trends in both nitrogen assimilation and phosphonate (C-P bond) acquisition and degradation (e.g., *phn* operon) systems in denitrifying specialists (**Supplementary Fig. 8; Supplementary Table 7**) were consistent with a genomic signature of adaptation to nutrient-rich (i.e., excess nitrogen and phosphorus) environments.”

Line 285: marginally significant? Isn't $p=0.092$ non-significant?

Response: We thank the reviewer for this methodological question. We used the term “marginally significant” for P -values between 0.05 and 0.1 based on a pre-defined analytical rationale, which were commonly used in scientific publications (Hsin et al., 2014; Sánchez-Bermejo et al., 2025; Young et al., 2025). Our study investigated the potential ecological thresholds in microbial functional adaptation along a nutrient gradient, a context characterized by high biological variability and complex with non-linear dynamics. In such exploratory analyses, a balance must be struck between Type I and Type II errors. Using a stricter P -value of 0.05 increases the risk of missing a meaningful biological signal (Type II error). Therefore, we adopted P -value = 0.1 as a more sensitive threshold for identifying potential trends that warrant discussion and may be crucial for generating new hypotheses for future validation. We agree that these results should not be overstated and have carefully toned down the statements by deleting the “marginally significant” and using “trend” throughout the section. Please refer to lines 709-721 in the revised manuscript.

“Specifically, specialists showed a trend of reduced pathways for nitrate/nitrite transporters (BinaryPGLMM, $P = 0.092$, *nrtB/nasE/cynB*, K15577; $P = 0.092$, *nrtA/nasF/cynA*, K15576; $P = 0.103$, *nrtC/nasD*, K15578; **Supplementary Fig. 8; Supplementary Table 7**) and assimilatory nitrate/nitrite reductases (BinaryPGLMM, $P = 0.099$, *nasC/nasA*, K00372; $P = 0.103$, *nasE*, K26138; $P = 0.103$, *nasD*, K26139; **Supplementary Fig. 8; Supplementary Table 7**). Critically, phosphorus metabolism genes were depleted in denitrifying specialists, including *LHPP* (BinaryPGLMM, $P = 0.038$, phosphatase, K11725), *mdoB* ($P = 0.166$, phosphoglucomutase, K01002), and *phn* operon genes ($P = 0.175$, *phnG*, K06166; $P = 0.231$, *phnH*, K06165; $P = 0.231$, *phnI*, K06164; $P = 0.231$, *phnL*, K05780; **Supplementary Fig. 8; Supplementary Table 7**), indicating relaxed selection for phosphonate scavenging in phosphate-rich habitats, a putative key factor in specialist evolution.”

Hsin, Amy, and Yu Xie. (2014). Explaining Asian Americans' academic advantage over whites. *Proceedings of the National Academy of Sciences* 111.23: 8416-8421.

Castro Sánchez-Bermejo, Pablo, et al. (2025). Intraspecific and intraindividual trait variability decrease with tree richness in a subtropical tree biodiversity experiment. *Nature Communications* 16.1: 11009.

Young, Euan A., et al. (2025). Mothers facing greater environmental adversity experience increased costs of reproduction. *Science Advances* 11.45: eadz6422.

Line 303: again convergent evolution and HGT are different things.

Response: Explained in our response to an earlier comment.

Line 316- 320: this is a long shot, please tone down.

Response: We agree and have toned down the rephrasing in the mentioned text. Please refer to lines 888-893 in the revised manuscript.

“Here we provide evidence extending this population-level paradigm to a natural environmental stressor, suggesting that coral resilience to chronic nutrient pollution is hard to infer from diversity or abundance shifts in denitrifier genera, but rather driven by the association with specific, hyper-efficient sub-species denitrifying populations within the dominant genus *Ruegeria*.”

Line 380-395: this paragraph is extremely speculative and not based on the evidence provided by the data.

Response: We agree and have thoroughly revised this paragraph to ensure all statements are grounded on our findings with key changes outlined below.

We have 1) scoped our primary conclusion to explicitly refer to “adaptation to nitrate gradients” rather than “anthropogenic change”; 2) acknowledged that presence of denitrifying specialists is likely one of the key factors that contribute to coral survival in western Hong Kong waters given the complex natural reef environments; 3) reframed the discussion on probiotic therapy from a prescriptive statement to a perspective for future work; and 4) toned down the generalized claims (e.g., “is often hidden”) with “provides a compelling case for the population-level paradigm”. Please refer to lines 990-1032 in the revised manuscript.

“This finding demonstrates that **coral adaptation to nitrate gradients** could be mediated by association with specialized bacterial populations within a genus, a mechanism that would be masked by genus-level microbial profiling. Given the complex natural reef environments, **presence of denitrifying specialists is likely one of the key factors that contribute to coral survival in western Hong Kong waters under chronic nutrient pollution**. **Future work** should employ laboratory experiments involving microbiome manipulation and nutrient gradients to validate the functional efficacy of denitrifying specialists on coral holobiont biology. As such, our research underscores that the functional capacity of a microbiome exerts itself at the sub-species level, **providing a compelling case** for a population-resolved

framework to understand host-microbe interactions and ecosystem functioning in a changing ocean.”

Fig 1C: fix ‘coral ompartment’

Response: We thank the reviewer for this careful observation. We have corrected this typo by using “Coral compartment” in the revised **Fig. 1c**.

Fig 2F: add x-axis label

Response: We have added the x-axis “Sampling site” in the revised **Fig. 2e** with corresponding site names indicated at the bottom of each boxplot.

Reviewer #1 (Remarks on code availability):

The results seem reproducible.

Response: We thank the reviewer for the positive assessment of our code availability.

Reviewer #2 (Remarks to the Author):

Xiang et al present a comparative genomic analysis of coral-associated *Ruegeria* found across Hong Kong's coral reefs. Coral reefs in Hong Kong's western regions are subject to high nitrogen concentrations compared to reefs in eastern Hong Kong. Xiang et al show that *Ruegeria* populations differ across this geography, wherein isolates from western areas show markedly higher denitrification activity compared to those from central/eastern areas despite most isolates possessing the genetic machinery for denitrification. The manuscript focuses on *Ruegeria* because "The predominance of *Ruegeria* further confirms its suggested important role in coral-associated denitrification process" (line 127). The relative abundances quoted at line 126 appear to be averaged across all reefs, and Figure 1B indicates that other denitrifying genera such as *Pseudomonas*, *Jhaorihella* and *Bradyrhizobium* are more relatively abundant than *Ruegeria* at YTW site which has the highest concentration of nitrogen. I think it would be interesting to address the extent *Ruegeria* contributes to the adaptation/survival of its coral hosts in these high N environments.

Response: We thank the reviewer for this insightful comment and the opportunity for us to clarify the rationale for focusing on *Ruegeria*. The reviewer is correct that at the YTW site (i.e., highest DIN in waters), the *nirS* denitrifying bacterial community showed the co-dominance of several genera, i.e., *Pseudomonas* (~10%), *Jhaorihella* (~9%), *Bradyrhizobium* (~8%), and *Ruegeria* (~8%), with *Ruegeria* and *Pseudomonas* being equally predominant (~12% each) in denitrifying bacterial communities across seven sampling sites. Our decision to focus the genomic and phenotypic analysis on *Ruegeria* was based on the following complementary reasons.

1- Technical feasibility for population-resolved analysis: *Ruegeria* yielded a robust, diverse collection of isolates (419 genomes), enabling the high-resolution population genomics that is the central methodology of our study. This depth of analysis was not technically feasible for other genera.

2- Excellent model with population-level differentiation: Our study centres on the fact that bacterial adaptation occurs at the sub-species population level within a ubiquitous genus. *Ruegeria* serves as an ideal model genus to demonstrate this "population-level paradigm", building upon our previous research (Luo et al., 2021; Chu et al., 2021; Lin et al., 2022).

We agree that quantifying the relative contribution of *Ruegeria* and other abundant genera to holobiont fitness is a fascinating question. To address the reviewer's point, we have added a sentence to the Discussion acknowledging that while *Ruegeria* populations are putatively key players, the co-dominance of other denitrifying genera suggests a potential for niche partitioning within the holobiont's denitrification consortium. We also propose future work involving microbiome manipulation and nutrient gradients, which can directly address the extent that *Ruegeria* contributes to the adaptation/survival of its coral hosts in high nutrient environments. Please refer to lines 998-1029 and lines 996-998 in the revised manuscript.

"While our study identifies population-level specialization within denitrifying *Ruegeria* as a putatively key mechanism in supporting coral resilience under high nutrients, the co-dominance of other denitrifying genera (e.g., *Pseudomonas*,

Jhaorihella) suggests a potential for niche partitioning within the holobiont's denitrification consortium, which warrants future investigation.”

“Future work should employ laboratory experiments involving microbiome manipulation and nutrient gradients to validate the functional efficacy of denitrifying specialists on coral holobiont biology.”

Luo, D. et al. (2021) Population differentiation of Rhodobacteraceae along with coral compartments. *The ISME Journal* 15, 3286-3302.

Chu, X., Li, S., Wang, S., Luo, D. & Luo, H. (2021) Gene loss through pseudogenization contributes to the ecological diversification of a generalist *Roseobacter* lineage. *The ISME journal* 15, 489-502.

Lin, X., McNichol, J., Chu, X., Qian, Y. & Luo, H. (2022) Cryptic niche differentiation of novel sediment ecotypes of *Ruegeria pomeroyi* correlates with nitrate respiration. *Environmental Microbiology* 24, 390-403.

One recommendation would be to relate the MCs to species and/or strain thresholds eg average nucleotide identity or relative evolutionary distance. Is an MC comparable to a *Ruegeria* species or strain, and do these MCs represent novel or existing *Ruegeria* species? If I understand correctly, PopCOGenT is based on gene flows which could become complicated depending on gene transfer events and geographic isolation? The genomes are mostly referred to as populations throughout the manuscript but also as strains in places (eg line 465).

Response: Our use of populations (i.e., main clusters, MCs) explicitly intends to capture the ecologically relevant units (Arevalo et al., 2019), not conventional taxonomic ones. MC is a population-level unit, comprising multiple closely related isolates. It is conceptually distinct from “strain” or “species”. To anchor this in the standard metrics, we have calculated that whole-genome average nucleotide identity (ANI) similarity between MCs ranged from 86.85% to 96.95% and isolates within the same MC shared $98.91 \pm 0.84\%$ ANI similarity. Scripts and raw data for ANI analysis were deposited in the GitHub repository ([https://github.com/444thLiao/Denitrification_Ruegeria.](https://github.com/444thLiao/Denitrification_Ruegeria)) in the Code availability section. Phylogenomic tree analysis of our 419 newly sequenced *Ruegeria* genomes with the 26 *Ruegeria* reference genomes (**Fig. 1c**) shows that our MCs were phylogenetically distinct from the published *Ruegeria* species, suggesting that they likely represent novel, ecologically distinct populations within this genus.

As the reviewer notes, PopCOGenT defines populations based on networks of recent gene flow (homologous recombination), making it robust to geographic isolation but sensitive to horizontal gene transfer (HGT) at shared loci. This aligns with its “reverse ecology” premise (Arevalo et al., 2019) to identify ecologically cohesive units based on genetic connectivity, which is ideal for detecting fine-scale microbial adaptation at the population level in our study. We have consistently used “population” or “MC” for the evolutionarily cohesive groups, and “isolates” for our cultured isolates throughout the revised manuscript. ANI-related analysis has been incorporated in lines 303-305 in the revised manuscript.

Arevalo, P., VanInsberghe, D., Elsherbini, J., Gore, J., & Polz, M. F. (2019). A reverse ecology approach based on a biological definition of microbial populations. *Cell*, 178(4), 820-834.

“Genomes within the same *Ruegeria* MC showed an average nucleotide identity (ANI) of 98.91% ± 0.84%, exceeding the established operational species threshold of 95%. ANI values between *Ruegeria* MCs ranged from 86.85% to 96.95%.”

More generally, I feel that it would improve readability of the manuscript if authors can orient readers by briefly explaining aims and methodology before describing results. For example, this paragraph to the next (line 177) goes from population genomics to amplicon profiling which won't be immediately clear to non-specialist readers. I understand this is largely personal preference in writing styles, but would be great to see explanation of the rationale and methods such that continuity between sections is more obvious.

Response: Point taken. We have added the explanation of the rationale and methods at the beginning of each Result section. Please refer to lines 211-215, 233-235, 378-380, 686-691 of the revised manuscript. We hope these modifications will improve the manuscript clarity and continuity.

“To assess the overall structure of the coral-associated denitrifier community along nitrate gradients, we first performed genus-level profiling by conducting amplicon sequencing of a denitrification marker gene. This approach allowed us to test whether local water nitrate gradients shape the denitrifying community compositions at the taxonomic level of bacterial genera. ”

“Having characterized the overall denitrifier community in corals, we next cultured isolates belonging to the predominant genus *Ruegeria* with their denitrification potential genomically evaluated.”

“To determine whether the denitrifying specialists exhibit superior denitrification activity than non-specialists, we quantified their productions of nitrous oxide (N₂O) and dinitrogen gas (N₂) using ¹⁵N-stable isotope labelling.”

“We further performed comparative genomics to uncover genetic differences, i.e., putative adaptations, underlying the denitrifying specialist phenotype. We used Phylogenetic signal estimation (Phylosig) and Phylogenetic Generalized Linear Mixed Model for Binary data (BinaryPGLMM) analyses ⁴³ to identify orthologous genes differentially associated with specialist MCs, along with the gene tree reconciliation analysis to investigate their evolutionary history (**Supplementary Dataset 5**).”

Comments

Line 94: are the nitrate/nitrite/ammonia concentrations consistently high in western HK throughout the year, or do they spike during certain seasons e.g. increased runoff and discharge in wet seasons? Vice versa for eastern waters.

Response: We thank the reviewer for this insightful question regarding the seasonal nutrient fluxes in seawaters. To address the reviewer's question, we have analysed

publicly available water quality data from 2012-2022 for our study region (Data source: Hong Kong EPD Marine Water Quality Report). The attached figure illustrates a persistent cross-regional gradient, confirming that dissolved inorganic nitrogen (DIN) levels in western sites (YTW and PC) exceed those of eastern sites throughout the year (i.e, across both wet and dry seasons). We have now added water DIN levels throughout the year as well as in both dry and wet seasons in the Supplementary Table 1. We also added one sentence to clarify the seasonal stability of this west-east nutrient gradient in Hong Kong reefs. Please refer to lines 109-200 in the revised manuscript.

“Hong Kong’s reefs, with their strong west-to-east nitrate gradient year-round (i.e., during both dry and wet seasons) ³¹ (Fig. 1a; Supplementary Table 1), provide a unique natural experiment to test this.”

[Figure Redacted]

Fig. 1C: This information is in the figure legend but should be included in the methods too- the genome tree in fig 1C has 26 reference *Ruegeria* genomes which should be labelled to provide context to the tree. Indicate what species and accessions to identify them, do they fall within any of the MCs, i.e. do the MCs represent known *Ruegeria* species.

Response: We thank the reviewer for this suggestion. We have revised the manuscript as follows :

1- The inclusion of 26 referenced *Ruegeria* genomes used to construct the phylogenomic tree (Fig. 1c) were added in the Methods section. Please refer to lines 1111-1113 in the revised manuscript.

“Orthologous gene families were identified using OrthoFinder (v2.5.1) incorporating 419 *Ruegeria* genomes along with 26 referenced *Ruegeria* genomes retrieved from public databases (Supplementary Dataset 3).”

2- All 26 referenced *Ruegeria* genomes were labelled in the revised Fig. 1c, with corresponding species names and NCBI accession numbers provided in the Supplementary Dataset 3. Additionally, 26 referenced *Ruegeria* genomes were deposited in the Zenodo (<https://zenodo.org/records/18171395>). As shown in the

revised Fig. 1c, 24 of 26 referenced *Ruegeria* genomes formed distinct clades by not falling within any of the MCs identified in our study. Only two referenced *Ruegeria conchae* strains clustered together with the MC6.

Line 165: It is not clear what data was used to construct the phylogenetic trees in figure 2C. On first read I thought it was based on *ATP5B*, *parC* and *nirS* genes from the isolate genomes to demonstrate that the three genes have sufficient resolution to distinguish between the 29 main clusters. The figure legend, however, says that trees are “based on concatenated single-copy orthologous genes”. At line 173, it is stated “phylogenetic trees constructed from amplified regions of all three genes...”. Can authors clarify what these trees are based on and generally edit this section to make it clear what was done.

Response: We confirm that the phylogenetic trees shown in Fig. 2c were indeed constructed from the designated amplicon of the *ATP5B*, *parC*, and *nirS* genes. These trees were essential for validating the capacity of the amplicons to distinguish between the 29 populations (MCs). We apologize for the mistaken texts in the previous figure legend, which has been corrected as “...based on the amplified region of each gene at the nucleotide level...” in the revised manuscript (line 1571).

Line 186: I think it would help readability if authors add slightly more description here as the current text suggests that envfit identified MCs in amplicon data. Maybe a brief line before this to indicate that the ASVs were assigned to MCs identified from

the genome tree by aligning ASVs to ... , followed by “envfit on the amplicon profiles showed that several MCs were enriched in high-nitrate waters.”

Response: Done. Please refer to lines 348-349 in the revised manuscript.

“ASVs were assigned to their respective MCs by aligning ASVs to the corresponding gene sequences extracted from *Ruegeria* genomes.”

Line 194: This statement cannot be verified as MC7 and MC40 are not shown in Fig. 2F. Maybe include a plot showing breakdown of the community by all MCs in addition to the four specific MCs in fig 2F. It would be interesting to know which MCs are more abundant in the low nitrate eastern waters.

Response: Excellent point. Compared to the compositional barplot merely showing the relative abundance of each MC based on the mean value, box plots with scatter points are advantageous on transparently showing the raw data. Showing all MCs in the Fig. 2e is not feasible as this would occupy too much space. To address this, we have added another supplementary figure (Supplementary Fig. 2 in the revised manuscript) showing the relative abundances of all MCs (including MC7 and MC40) in coral-associated *Ruegeria* communities across sites.

As shown in Supplementary Fig. 2 in the revised manuscript, MC50, MC33, MC25 and MC10 were more abundant in the low nitrate eastern waters compared to western ones. Given that these MCs each constituted <1% of the total *Ruegeria* populations, below our abundance threshold (>1%) considered for an ecologically meaningful proportion, we did not include them in the denitrification activity measurements.

Line 226: “thus the strains affiliated with specialist MCs were not necessarily isolated from the western sites only”. Correct me if I am mistaken- since MCs are derived from phylogenetic clusters on the genome tree (figure 1C), it is possible that within each MC their nirS sequence (or other genes) can differ- this is shown in fig 2C where the MCs are not entirely monophyletic in the gene trees. As such could there be cases where isolates from for example MC6 may not necessarily be a “specialist denitrifier” as their copy of denitrification genes could be a “low efficiency” version?

Response: We sincerely thank the reviewer for raising this insightful question regarding the genetic and functional consistency of isolates within a denitrifying specialist main cluster (MC). The reviewer is correct in noting that, within an MC,

individual isolates may harbor divergent copies of denitrification genes (as suggested by the non-monophyletic patterns in the *nirS* gene trees). This is an important observation that touches upon the core principles of our population-resolved framework. Below, we clarify the conceptual and methodological foundations of our approach, which we believe will resolve the reviewer's query.

1- PopCOGenT defines populations based on recent gene flow, not phylogeny. MCs are not delineated from a phylogenomic tree (**Fig. 1c**). Instead, PopCOGenT analyzes recent gene flow (homologous recombination) across entire genomes to identify clusters of isolates that form an ecologically cohesive and genetically connected unit (Arevalo et al., 2019). Therefore, the population boundary is defined by contemporary gene exchange and ecological coherence, not by phylogenetic distance or the evolutionary history of any single gene.

2- The “denitrifying specialist” is assigned at the population level based on the environmental enrichment. We defined a given MC as a “denitrifying specialist” based on a statistically robust, population-wide pattern: the MC as a whole showed significant enrichment in corals from high-nitrate western sites in our environmental metabarcoding data (**Fig. 2d-f**). This environmental signature reflects the ecological prevalence and putative adaptive character of the population under high-nitrate conditions. Consequently, all isolates belonging to such MC are termed “specialist denitrifiers” because they are members of a population that is ecologically associated with high-nitrate niches.

3- Functional variation within a specialist population is expected and does not contradict its classification. The reviewer correctly hypothesizes that individual isolates within a specialist MC (e.g., MC6) could possess “low-efficiency” variants of denitrification genes. This is entirely plausible and aligns with population-genetic theory. Members of a specialist population can migrate to lower-nitrate sites (e.g., eastern reefs), where selective pressure for high denitrification efficiency may be relaxed. Over time, genetic drift or local adaptation could lead to reduced efficiency in some individuals, yet these isolates remain part of the same gene-flow unit (i.e., the same population) and retain the genomic backbone that defines the MC. Our phenotypic assays captured the population-level trend: isolates from specialist MCs exhibited significantly higher denitrification rates *on average*, with individual variation being present (**Fig. 3bc**).

4- Our study design accounts for this variation. In our stable-isotope assays, we intentionally included isolates from both western and non-western sites within specialist MCs. This allowed us to test whether the specialist phenotype was robust across the population's geographic range. The significant phylogenetic ANOVA ($P < 0.05$) confirmed that, despite some intra-population variation, specialist MCs as a group outperformed non-specialists. This result underscores that the adaptive trait (high denitrification activity) is a property of the population, even if not uniformly present in every member.

In summary, the reviewer's comment highlights an important nuance: population-level ecological specialization does not require every individual isolate to be phenotypically identical. PopCOGenT identifies ecologically relevant populations based on genome-wide gene flow, and our environmental data show that these

populations are consistently associated with high-nitrate reefs. Functional variation within a population is both expected and informative, reflecting the dynamic interplay between gene flow, migration, and local selection. We have added a brief clarification (lines 912-914) in the revised manuscript to emphasize that specialist MCs are defined by their collective environmental enrichment, and that intra-population functional diversity can arise through migration and relaxed selection.

“Specialist MCs are defined by their collective environmental enrichment in high nitrate waters with intra-population functional differences that can arise through migration and relaxed selection. Members of a specialist population can migrate to low-nitrate sites (e.g., eastern reefs), where selective pressure for high denitrification activity may be relaxed. Over time, genetic drift or local adaptation could lead to reduced denitrification efficiency in some individuals, yet these isolates remain part of the same gene-flow unit (i.e., the same population) and retain the genomic backbone that defines the MC.”

Arevalo, P., VanInsberghe, D., Elsherbini, J., Gore, J., & Polz, M. F. (2019). A reverse ecology approach based on a biological definition of microbial populations. *Cell*, 178(4), 820-834.

Related to above: how do these MCs relate to conventional species thresholds- are the MC members “strains” (i.e. same species) as authors refer to in the manuscript, or are they different species of *Ruegeria*? Maybe check ANI or GTDB relative evolutionary divergence thresholds.

Response: Thank you for raising this important point. In our study, “MCs” are defined by PopCOGenT (Arevalo et al., 2019), which captures recent gene flow and provides a finer resolution of ecologically relevant units than traditional species delineation boundaries (Parks et al., 2020). MC is a population-level unit, comprising multiple closely related isolates and is conceptually distinct from “strain” or “species”. *Ruegeria* genomes within the MC show the whole-genome average nucleotide identity (ANI) values of $98.91\% \pm 0.84\%$, exceeding the established species threshold of 95%, which is also used by GTDB (Parks et al., 2020; Rodriguez-R et al., 2024). GTDB incorporates the relative evolutionary divergence (RED) as a complementary metric. When comparing genomes across MCs, the highest pair-wise ANI value is $91.9\% \pm 5.05\%$, indicating that many MCs correspond to different species under the established ANI-based species threshold. Scripts and raw data for ANI analysis were deposited in our repository (https://github.com/444thLiao/Denitrification_Ruegeria.) in the Code availability section. We have added this relevant information in lines 303-305 in the revised manuscript.

Arevalo, P., VanInsberghe, D., Elsherbini, J., Gore, J., & Polz, M. F. (2019). A reverse ecology approach based on a biological definition of microbial populations. *Cell*, 178(4), 820-834.

Parks D.H. et al. (2020). A complete domain-to-species taxonomy for bacteria and archaea. *Nat. Biotechnol.*, 38, 1079–1086.

Rodriguez-R, L. M., Conrad, R. E., Viver, T., Feistel, D. J., Lindner, B. G., Venter, S. N., ... & Konstantinidis, K. T. (2024). An ANI gap within bacterial species that advances the definitions of intra-species units. *MBio*, 15(1), e02696-23.

“Genomes within the same *Ruegeria* MC showed an average nucleotide identity (ANI) of 98.91% ± 0.84%, exceeding the established operational species threshold of 95%. ANI values between *Ruegeria* MCs ranged from 86.85% to 96.95%.”

Line 340: “Denitrifying specialist strains MC1-AF0 and MC6-O11 achieved exceptional N₂ production rates (~480 and 210 pmol N₂ ml⁻¹ h⁻¹, respectively), surpassing a known denitrifying *Methylomirabilis* bacterium by three orders of magnitudes under a comparable 15N-labeling condition (estimated from 1.5 μmol 30N₂ productions by 50 mg wet biomass (~1010 CFU) in 150 ml anoxic serum vial over four hours).” I feel that this comparison could come across as disingenuous since authors claim that large phenotypic variability exists within *Ruegeria* populations, and as such the same logic should also apply to *Methylomirabilis* (i.e. does this specific *Methylomirabilis* species/strain have the highest denitrifying activity among other *Methylomirabilis*), and why compare *Ruegeria* with this *Methylomirabilis* isolated from paddy soils? It would be more relevant if authors compared with other denitrifiers in corals instead, eg *Pseudomonas*, *Jhaorihella* which were more relatively abundant than *Ruegeria* in YTW.

Response: We agree that comparing coral-associated *Ruegeria* to a soil denitrifier *Methylomirabilis* is ecologically inappropriate. Our intent was merely to use this study as a methodological benchmark, given the lack of published ¹⁵N-stable isotope assay data for coral-associated denitrifying isolates like *Pseudomonas* or *Jhaorihella*. We have revised the manuscript as follows. Please refer to lines 921-953 in the revised manuscript.

- 1- Deleted the inappropriate quantitative comparison statement.
- 2- Rewrote the sentence to highlight our methodological novelty and intrinsic findings.

“To our knowledge, this study pioneered the quantification of complete denitrification rates (considering both N₂O and N₂) for pure denitrifier cultures isolated from corals using the ¹⁵N-stable isotope assay. Direct and simultaneous quantification of both ¹⁵N-labelled N₂O and N₂ ⁵⁰ represents a methodological advancement over traditional approaches that infer denitrification rates solely from N₂O accumulations ^{51,52}, enabling accurate resolution of denitrification activity in specialist and non-specialist MCs.”

Minor comments

Line 49: superior efficiency compared to?

Response: Addressed in line 49 in the revised manuscript.

“.... compared to non-specialists.”

I suggest removing filler text eg “vibrant bastions of marine diversity” (line 56),

“insidious” (line 59)

Response: Addressed (lines 61) in the revised manuscript.

Line 63: it wasn't immediately clear what algae is referred to since nitrogen pollution also stimulates macroalgal proliferation. Maybe Symbiodinium algae?

Response: We have replaced “algal” with “**symbiotic algal Symbiodiniaceae**” in lines 66-67 in the revised manuscript.

Line 91: perhaps a typo – eighty coral species “obstructing” the reefs?

Response: We have replaced the typo by “**inhabiting**” in the revised manuscript (line 97).

Line 102: not clear whether “fine-scale population (i.e. sub-species level)” refers to coral or microbial populations.

Response: It refers to the “**microbial population**”, which has been updated in line 108 in the revised manuscript.

Figure 1A: typo “Ammnoia”

Response: Corrected.

Figure 1C: typo “Coral ompartment”. Maybe also indicate in the legend criteria for defining main clusters, e.g. 95% ANI?

Response: Corrected and added the criteria for MC definition in the revised manuscript (line 1537).

“**MCs were delineated using PopCOGenT.**”

Line 150: I would suggest changing “detoxifying eutrophic reefs”- detoxifying does not sound appropriate in this context.

Response: We have replaced it with “**removing excess nitrate/nitrite for corals in eutrophic reefs**” in the revised manuscript (lines 285-288).

Line 516: what is “mb”?

Response: It was a typo that we have deleted in the revised manuscript (line 1228).

Line 183: This sentence needs editing “This stark contrast between the site-specific functional community structure and the stable genus-level abundance, underscoring the critical population dynamics we aimed to uncover.”

Response: We have updated this sentence in lines 323-348 in the revised manuscript.

“This stark contrast between the site-specific denitrifying *Ruegeria* community composition and the evenly distributed denitrifying genera across sites (Fig. 1b; Fig. 2d) highlights the critical population dynamics we aimed to uncover.”

Line 194: Fig. 2F is referenced in text before 2E, suggest swapping panels.

Response: Done.

Line 276, 285, 288, 296: typo “BinayPGLMM”

Response: Corrected throughout the revised manuscript.

Line 424- I believe should be “recipe” not “receipt”

Response: Corrected (line 1061 in the revised manuscript).

Line 478: Not sure what is meant by “17 MCs included per closed genome per MC”

Response: We have rephrased it in lines 1134-1135 in the revised manuscript.

“Each MC contained one representative closed genome for the remaining 17 MCs.”

Line 514: “The phylogenetic trees showing the ability to resolve different *Ruegeria* MCs were kept for the primer design”. Was there a measure of congruence between the gene (fig2C) and genome tree (Fig 1C)? I believe some of the MCs in fig2C are polyphyletic. Maybe label the MCs in fig2C as some of the colours are difficult to tell apart, and I think some MC colours do not appear in the tree in Fig1C? Do the colours in Fig1C and all panels in fig2C represent the same MCs?

Response: We appreciate the reviewer’s insightful questions about the phylogenetic trees. We followed a population genetics framework, where a “population” is defined as a group of individuals of the same species that exchange genetic material and

inhabit a specific geographic area, thereby sharing a common gene pool. The PopCOGenT tool delineates such populations by identifying clusters of recent gene flow, which represent ecologically cohesive and genetically isolated units. As such, MC is independent of the genome tree. Regarding the congruence, the trees in Fig. 1c (whole-genome phylogeny) and Fig. 2c (single-gene trees) serve different purposes and are not expected to be perfectly congruent. The genome tree displays the ecological niches and denitrification gene presence of MCs, while the gene trees focus on demonstrating the utility of our population-resolving gene markers. Discordance between Fig. 1c and Fig. 2c is common due to factors such as incomplete lineage sorting or differential horizontal gene transfer at specific loci. Regarding the figure clarity, all MCs in Fig. 2c are now clearly labelled. We have also ensured that the same color consistently represents the same MC across both Fig. 1c and Fig. 2c. We hope these changes will improve the clarity of our figures.

Reviewer #3 (Remarks to the Author):

In this manuscript Xiang et al. examine the role of denitrifying microbial partners within the coral microbiome and their contribution to resilience in the face of increasing eutrophication along Hong Kong's coral reef. The study sets out to understand why some reefs in high concentration nitrate/nitrite areas exhibit resilience and resist dysbiosis, which would otherwise be expected from the excess nitrogen leading to the proliferation of Symbiodiniaceae. The authors conclude that it is due to the presence of dominant denitrifying genera like *Ruegeria*. *Ruegeria*, through genomic inquiry, is demonstrated to have the genetic capability to fully reduce nitrate and nitrite down to N₂, which is not bioavailable for assimilation. Furthermore, although *Ruegeria* is ubiquitous across the nitrogen gradient of the Hong Kong Bay, it is demonstrated that specific strains are much more adept at this reduction and are able to execute it with upwards of 10-fold efficiency than others. There is a selective association of these efficient reducers with coral colonies in high nitrogen areas and the authors put forward that it is this metabolic process that curbs dysbiosis by limiting bioavailable nitrogen sources to the coral colony and microbiome.

Line 60: The clause “another insidious threat namely nutrient pollution, is increasingly” is repeated.

Response: We thank the reviewer for the careful observation. We have deleted the repeated sentence in the revised manuscript (lines 63-64).

Line 84: Is it known which sections of denitrification gene clusters need to be intact within a genome for functional expression of that process?

Response: Thank you for this insightful question. Yes, for a bacterium to perform complete denitrification (NO₃⁻/NO₂⁻ to N₂), it is well-established that a core set of gene clusters must be intact and expressed, including *narGHJI* or *napAB* operons for NO₃⁻ reduction, *nirS* or *nirK* operons for NO₂⁻ reduction, *nor* operons for N₂O production, and *nos* operons for N₂ production (Zumft, 1997; Lycus et al., 2017). Guided by this principle, we systematically assessed the presence/absence of core denitrification genes across 419 newly sequenced *Ruegeria* genomes (**Figure 1c; Supplementary Dataset 2**).

Zumft, W. G. (1997). Cell biology and molecular basis of denitrification. *Microbiology and molecular biology reviews*, 61(4), 533-616.

Lycus, P., Lovise Bøthun, K., Bergaust, L., Peele Shapleigh, J., Reier Bakken, L., & Frostegård, Å. (2017). Phenotypic and genotypic richness of denitrifiers revealed by a novel isolation strategy. *The ISME Journal*, 11(10), 2219-2232.

Line 106: Have the authors investigated the dispersion of the ubiquitous bacteria in the water column across the area comprising the nitrate gradient? If there is equal dispersion, it would point more towards coral animal selectivity of denitrifying bacteria. If denitrifying bacteria are more concentrated in areas where they can make a better living, however, them ending up within the microbiomes of the corals there may be a byproduct of free association.

Response: It is an excellent point. We acknowledge that our study did not survey microbial communities in the water column across the nitrate gradients. Distinguishing between proactive and passive associations of denitrifying specialists in corals requires paired sampling of corals and their immediate surrounding water to draw meaningful conclusions. Our field sampling in the year 2022 was designed to elucidate the *in situ* association and functional capacity of coral-associated microbes and did not include ambient water collection. We apologize that a sampling now in the year 2026 would not yield the required paired environmental data.

While distinguishing between proactive host selection and passive environmental association is a compelling question, it is beyond the scope of our current study. To ensure our interpretations are precisely aligned with our findings, we have revised the manuscript to use the neutral term “association” when describing the presence of specialist MCs in corals, replacing terms like “selective enrichment” that imply a specific mechanism. We fully agree that paired environmental sampling is crucial and will incorporate this design, including simultaneous sampling of corals and surrounding environmental substrates (i.e., seawater and sediments), into our future work to unravel the ecological mechanism.

Line 121: “The *nirS* amplicon analysis revealed that denitrifier abundance in corals were evenly distributed at the genus level along water nitrate gradients.” To my last point, this was determined through analysis of their presence in the coral mucus, tissue, and skeleton, correct?

Response: Yes, the reviewer is correct. The *nirS* amplicon analysis was performed on merged sequencing results from all coral holobiont members, including mucus, tissue, and the skeleton. This approach was taken to profile the integrated, coral-associated denitrifier community as a functional unit. We have clarified it in lines 221-222 (Result) and lines 1259-1260 (Method) in the revised manuscript.

“The *nirS* amplicon analysis revealed that denitrifier abundances in corals (i.e., using combined sequenced compartments of mucus, tissue, and the skeleton) were evenly distributed at the genus level across the nitrate gradient...”

“DNA extraction for coral samples (i.e., mucus, tissue, and skeletal compartments) was performed with the DNeasy PowerSoil Pro Kits (Qiagen, Germany)...”

Line 127: “The predominance of *Ruegeria* further confirms...” Does the relative stability of *Ruegeria*’s proportionality within populations across the nitrate gradient complicate the author’s point that it aids resilience for corals at the high end of the gradient? How does this correlate to nitrate-driven selection?

Response: We thank the reviewer for raising this crucial point. The reviewer is correct to note the apparent paradox: the genus-level stability of *Ruegeria* proportion seems at odds with its proposed role in site-specific enrichments. This observation is what motivated our investigation on the sub-genus population level. The dominant denitrifier *Ruegeria* serves as an excellent model as our prior works have shown profound intra-taxon functional heterogeneity and niche differentiation within this genus, where ecologically relevant traits vary at the population level (Luo et al., 2021; Chu et al., 2021; Lin et al., 2022). As a result, our study highlights that nitrate-driven

association operates not between genera, but within sub-species populations. The stable *Ruegeria* genus proportion across nitrate gradients is the “background” against which the critical signal emerges: the association of specific, hyper-efficient denitrifying *Ruegeria* specialists in high-nitrate water corals (**Fig. 2d-f**).

Luo, D. et al. (2021). Population differentiation of Rhodobacteraceae along with coral compartments. *The ISME Journal* 15, 3286-3302.

Chu, X. et al. (2021). Gene loss through pseudogenization contributes to the ecological diversification of a generalist *Roseobacter* lineage. *The ISME journal* 15, 489-502.

Lin, X. et al. (2022). Cryptic niche differentiation of novel sediment ecotypes of *Ruegeria pomeroyi* correlates with nitrate respiration. *Environmental Microbiology* 24, 390-403.

Line 145: I see that last point gets picked up here!

Response: Yes, we indeed built a logistic flow from the observed evenly distributed denitrifying genus abundance to the subsequent step of decoding the enrichment of denitrifying specialists at the population level.

Line 155: This is an apt explanation of what was done although, as written, it might be more appropriate in the methods section.

Response: We thank the reviewer for this suggestion. We have carefully considered and preferred to keep the original sentence here, as it provides a logical flow to introduce our core population genomic framework.

Line 209: “Denitrifying specialist *Ruegeria* MCs exhibited significantly higher productions...” belies statistical comparisons that were conducted between production levels. Those values should be included. Also, the text says that the denitrifying specialists were compared to the non-specialists, which suggests a comparison of two groups. The previous sentence states that four specialist MCs and nine non-specialist MCs (13 total) were being considered. It is slightly confusing here if the MCs are considered independently or as a group. In that case that they are considered as a group, can the authors comment on their preference for an ANOVA, which is typically used for comparing three or more groups. If, however, more than two MCs are being compared independently across phylogenetic lineage, that should be made clearer in the text.

Response: We thank the reviewer for these critical points, which allow us to clarify our statistical approach and improve the manuscript’s clarity. The Phylogenetic ANOVA (PhylANOVA) model included two fixed factors with denitrification trait (i.e., specialists vs. non-specialists) as the primary predictor and MC identity as a covariate. In this light, more than two MCs are being compared independently across phylogenetic lineage. The independent statistical units are the two pre-defined ecological groups: “Denitrifying Specialists” (n = 4 MCs) and “Non-specialists” (n = 9 MCs). We used to PhylANOVA control for the phylogenetic non-independence of

isolates within one denitrification trait group. Since the isolates shared the evolutionary history (Fig. 3a), a standard ANOVA would violate the assumption of data independence, inflating the risk of Type I error. PhylANOVA incorporates the phylogeny, which can provide a statistically robust comparison of group means, even when comparing two groups. This method is established for such comparisons in evolutionary biology and ecology (Losos et al., 1998; Manthey et al., 2016; Adams et al., 2019).

Losos, J. B., Jackman, T. R., Larson, A., Queiroz, K. D., & Rodríguez-Schettino, L. (1998). Contingency and determinism in replicated adaptive radiations of island lizards. *Science*, 279(5359), 2115-2118.

Manthey, J. D., Campillo, L. C., Burns, K. J., & Moyle, R. G. (2016). Comparison of target-capture and restriction-site associated DNA sequencing for phylogenomics: a test in cardinalid tanagers (Aves, Genus: *Piranga*). *Systematic biology*, 65(4), 640-650.

Adams, D. C., & Collyer, M. L. (2019). Phylogenetic comparative methods and the evolution of multivariate phenotypes. *Annual Review of Ecology, Evolution, and Systematics*, 50(1), 405-425.

Line 210: $^{46}\text{N}_2\text{O}$ would more accurately be notated as $^{15}\text{N}_2\text{O}$ to denote that it is the stable isotope version of N_2O being investigated for the calculation. Similarly, for $^{30}\text{N}_2$

Response: We thank the reviewer for this point. We have carefully considered this and propose to retain the notations $^{46}\text{N}_2\text{O}$ and $^{30}\text{N}_2$ in our revised manuscript following the below rationale.

Our study uses the ^{15}N Nitrogen Isotope Pairing Technique (IPT), which requires distinguishing between singly- and doubly- ^{15}N -labeled gas products (e.g., $^{45}\text{N}_2\text{O}$ vs. $^{46}\text{N}_2\text{O}$, $^{29}\text{N}_2$ vs. $^{30}\text{N}_2$) to calculate the denitrification rates. The notations $^{46}\text{N}_2\text{O}$ and $^{30}\text{N}_2$ specify the doubly-labelled ^{15}N molecules, which are the direct end-products of complete denitrification from our added $\text{Na}^{15}\text{NO}_3$ tracer. In contrast, using $^{15}\text{N}_2\text{O}$ or $^{15}\text{N}_2$ could be obscure whether the data refers to the singly- or doubly-labelled products. Our stable isotopic data report fits the principle of the IPT methodology and is consistently used in high-profile research (Babbin et al., 2014; Wan et al., 2023; Su et al., 2022; Xu et al., 2020).

Babbin, A. R., Keil, R. G., Devol, A. H., & Ward, B. B. (2014). Organic matter stoichiometry, flux, and oxygen control nitrogen loss in the ocean. *Science*, 344(6182), 406-408.

Wan, X. S., Sheng, H. X., Liu, L., Shen, H., Tang, W., Zou, W., ... & Kao, S. J. (2023). Particle-associated denitrification is the primary source of N_2O in oxic coastal waters. *Nature Communications*, 14(1), 8280.

Su, X., Yang, L., Yang, K., Tang, Y., Wen, T., Wang, Y., ... & Zhu, Y. G. (2022). Estuarine plastisphere as an overlooked source of N₂O production. *Nature Communications*, 13(1), 3884.

Xu, C., Zhang, K., Zhu, W., Xiao, J., Zhu, C., Zhang, N., ... & Cheng, L. (2020). Large losses of ammonium-nitrogen from a rice ecosystem under elevated CO₂. *Science Advances*, 6(42), eabb7433.

Line 253: Extended Data Figs 4 & 5, numerous labels overlap making the annotations illegible. Consider offsetting some of the labels.

Response: We have followed your suggestions on re-organizing the labels in the Extended Data Fig. 4 (Supplementary Fig. 6 now) and Fig. 5 (Supplementary Fig. 7 now). We have ensured that there is no overlap between labels/text annotations in the revised figures.

Line 348: “Quantification of end-product...specialist MCs.” This is alluding to non-specialist MCs lacking the ability to fully reduce to N₂, correct?

Response: We agree that the original sentence was vague and could be misinterpreted. To clarify, both denitrifying specialist and non-specialist MCs possess the genetic potential (**Fig. 1c**) and phenotypic capacity (**Fig. 3bc**) to carry out a complete denitrification process with N₂ as the end product. The key difference between denitrifying specialists and non-specialists is of quantitative nature, i.e. denitrification activity. We have rephrased it in the revised manuscript (lines 950-953) to accurately reflect our findings.

“Direct and simultaneous quantification of both ¹⁵N-labelled N₂O and N₂ ⁵⁰ represents a methodological advancement over traditional approaches that infer denitrification rates solely from N₂O accumulations ^{51,52}, enabling accurate resolution of denitrification activity in specialist and non-specialist MCs..”

Line 384: Can the authors comment on if the “selective enrichment” within a dominant genus like *Ruegeria* is host derived? Furthermore, all things being equal, what might prevent a colony within a eutrophic area with a specialist MC population in the water column from being colonized by it? Why aren't the benefits of this advantageous population ubiquitous throughout the zone?

We thank the reviewer for these excellent questions, which motivate us to explore this deeper in our future works. Below, we have addressed each of your points based on our data and current knowledge.

1- “Whether the association is host-derived”: While our study cannot directly distinguish between host selection and environmental filtering, the correlation between specialist MCs and resilient coral host species in degraded, high-nitrate reefs suggests a potential host-mediated process. These resilient corals may possess traits that facilitate association with these hyper-efficient denitrifying specialists.

2- “Potential barriers to prevent colonization”: Even with denitrifying specialists present in the water column, ecological barriers at the coral genus or species level can

potentially prevent bacterial colonization. These barriers include occupied ecological niches by established stable microbiomes, as well as coral host-specific characters such as genetic variation, immune function, and mucus composition that may lead to differential colonization compatibility.

3- “Context-dependent distribution”: The specialist phenotype may not be universally advantageous. Its high-performance denitrification machinery may incur metabolic trade-offs (e.g., energy maintenance expenses) (Pold et al., 2025) that are only beneficial in high-nitrate niches.

Pold, G., Saghäi, A., Jones, C. M., & Hallin, S. (2025). Denitrification is a community trait with partial pathways dominating across microbial genomes and biomes. *Nature Communications*, 16(1), 9495.

Line 646: This is a strange way to end the sentence with reference to a publication. Perhaps you want to write, “...according to Yao et al.50”

Response: Point taken. We have added “according to Yao et al., (2024)” in line 1384 in the revised manuscript.

Line 679: MS data should be put into a repository, such as MassIVE, for the community.

Response: We thank the reviewer for this suggestion. Raw data files from the ¹⁵N-stable isotope assay have been deposited in the public repository Zenodo. The dataset is permanently accessible via the link (<https://zenodo.org/records/18028409>). We have added this text in lines 1686-1688 in the revised manuscript.

“Raw data from the ¹⁵N-stable isotope assays has been deposited in the public repository Zenodo (<https://zenodo.org/records/18028409>).”

Line 813: It is not immediately clear what statistical comparisons are being made in panels B & C. I recommend removing the legend text referencing isolate site and MC from the main plot field of panel B, that would then leave the field open to clearly denote what comparisons from the applicable results section are being referenced to in the plots.

Response: We thank the reviewer for this excellent suggestion on improving the clarity of our statistical comparisons in the **Fig. 3b and 3c**. We have removed the legend texts and denoted the statistical comparison groups described in the revised Result section.

General note on the figure scale, I assume this will be adjusted when the manuscript goes to print? They are quite large and not formatted for a printed page, which makes them difficult to consume and reference in the course of reviewing, although they are generally visually appealing.

Response: We have carefully adjusted the scale, layout, and font sizes of all figures to comply with journal's formatting guidelines for publication, ensuring they are clear in the final printed and online formats.

Reviewer #3 (Remarks on code availability):

I think the README is sufficient for the genomic data.

Response: We thank the reviewer for the positive assessment of our code availability.

Decoding Coral Resistance to Eutrophication: Association of Hyper-Efficient Denitrifiers as Key Microbial Allies

We would like to thank the Nature Communications editors for accepting our manuscript. We have provided detailed point-by-point responses to each of the reviewer's points in **blue** and highlighted changes to the manuscript in **green** (which are also pointed out in the track-changes version of the manuscript file). Please note that the line numbers cited in our responses refer to the clean version of the revised manuscript.

REVIEWERS' COMMENTS

Reviewer #1 (Remarks to the Author):

I commend the authors for the improvement to the manuscript. I have a few more comments that I would like to see addressed before publication and that I think would improve the reach of this study as well as its relevance to the coral microbiology literature.

In the discussion I suggest that the authors articulate the involvement of skeleton-dwelling microbes in the coral holobiont metabolism. Often, skeleton-dwellers are not considered important to the holobiont physiology, for instance, this recent piece (<https://doi.org/10.3390/microorganisms14010202>) suggest that tissue associated microbes should be targeted for future probiotic strategies. Instead, the present study proves that skeleton-dwellers have important roles in the overall holobiont physiology and this should be articulated.

Response: Thanks for this suggestion. We have added two sentences in the discussion part to highlight the importance of skeleton-dwelling microbes in the coral holobiont metabolism. Please refer to lines 408-412 in the revised manuscript.

“While a recent review proposed targeting tissue-associated microbes for probiotic interventions⁵³, our denitrifying specialist isolates were predominantly obtained from the skeleton of *Oulastrea crispata*. This reveals an importance of often-overlooked skeleton-dwelling microbes in coral metabolism”.

Line numbers referring to the clean version of the manuscript.

53-54: is this sentence complete? It reads awkwardly.

Response: The sentence has been revised for clarity. Please refer to lines 51-54 in the revised manuscript.

“As such, our work resolves the paradox of why dominant denitrifying genera are ubiquitous, yet only certain corals thrive in eutrophic conditions. It also provides a framework for future studies delineating ecologically important host-associated microbes”.

138-139: I suggest changing the title to “A majority of coral-associated *Ruegeria* isolates carry a complete denitrification gene set”

Response: Done. Please refer to lines 138-139 in the revised manuscript.

“A majority of coral-associated *Ruegeria* isolates carry a complete denitrification gene set”

155-156: what was the completeness of the genomes that possessed a partial denitrification gene set or lacked denitrification genes? Could it be that they didn't have these genes because of sequencing bias? I think this point should be made explicit in the text.

Response: We thank the reviewer for raising this important point. To address this concern, we have provided the genome completeness of all 419 *Ruegeria* isolates with statistical support between isolates carrying complete versus incomplete or no denitrification genes. Please refer to lines 155-160 in the revised manuscript.

Genome completeness and contamination information have been also added to the revised Dataset S2.

“Genome completeness was uniformly high across all isolates (98.59 ± 1.31 %), with no significant difference (Mann-Whitney U test, $P = 0.901$) between isolates carrying complete (98.58 ± 1.41 %) versus partial or no denitrification genes (98.66 ± 0.55 %), confirming that this variation reflects biological differences rather than sequencing or assembly artifacts”.

Reviewer #1 (Remarks on code availability):

The README file is fine

Reviewer #2 (Remarks to the Author):

I thank the authors for addressing all reviewer comments, and for explaining the concept of main clusters (MCs) used in this manuscript. One final comment I have is regarding the use of the term “sub-species groups” when referring to the MCs. As authors pointed out that since MCs are distinct from the conventional species or strain level taxonomic ranks, I wonder if it still makes sense to refer to the MCs as sub-species populations (e.g. at lines 280, 299, 892, and 901 of the revised manuscript)? I would think “... driven by the association with specific, hyper-efficient denitrifying populations within the dominant genus *Ruegeria*” (line 892) or “capacity for associations of microbial allies at a MC population level” (line 901) would be more accurate in these contexts.

Response: Excellent point. We have replaced “sub-species groups” with “sub-genus groups” throughout the revised manuscript to avoid any confusion.

Reviewer #3 (Remarks to the Author):

I have considered the authors review letter with their responses to my and my colleagues concerns from the initial review and the updated manuscript. I find my concerns to have been adequately satisfied.

I would implore the authors to carefully review the figures. Panel 1A has location markers that obscure the location text. Panels 3N and 3C have ill spaced facets that are not centered. Prior to publication these noticeable details should be fixed to present a polished final product to the community.

Beyond that, I have no further suggestions at this time.

Response: We thank the reviewer for their careful attention to the figures. We have thoroughly revised the figures to address the following raised points:

Fig. 1a: The location markers have been resized and repositioned to ensure they no longer obscure the location text. Map labels are now clearly legible.

Fig. 3b and 3c: The facet spacing has been corrected, and all panels are now properly aligned and centred. We have standardized the dimensions across facets to ensure visual consistency.

We have also conducted a full review of all remaining figure panels to ensure similar formatting issues are not present elsewhere.